# Diatom heterotrophy on brown algal polysaccharides emerged through horizontal gene transfer, gene duplication, and neofunctionalization

Zeng Hao Lim[1,2], Peng Zheng[1], Christopher Quek[1], Minou Nowrousian[3], Finn L. Aachmann[4], Gregory Jedd[1,2]*

1 Temasek Life Sciences Laboratory, Singapore, Singapore, 2 Department of Biological Sciences, National University of Singapore, Singapore, Singapore, 3 Department of Molecular and Cellular Botany, Ruhr-Universität Bochum, Bochum, Germany, 4 Norwegian Biopolymer Laboratory (NOBIPOL), Department of Biotechnology and Food Science, Norwegian University of Science and Technology (NTNU), Trondheim, Norway

* gregory@tll.org.sg

## Abstract

A major goal of evolutionary biology is to identify the genetic basis for the emergence of complex adaptive traits. Diatoms are ancestrally photosynthetic microalgae. However, in the genus *Nitzschia*, loss of photosynthesis led to a group of free-living secondary heterotrophs whose manner of acquiring chemical energy is unclear. Here, we sequence the genome of the non-photosynthetic diatom *Nitzschia* sing1 and identify the genetic basis for its catabolism of the brown algal cell wall polysaccharide alginate. *N*. sing1 obtained an endolytic alginate lyase enzyme by horizontal gene transfer (HGT) from a marine bacterium. Subsequent gene duplication through unequal crossing over and transposition led to 91 genes in three distinct gene families. One family retains the ancestral endolytic enzyme function. By contrast, the two others underwent domain duplication, gain, loss, rearrangement, and mutation to encode novel functions that can account for oligosaccharide import through the endomembrane system and the exolytic production of alginate monosaccharides. Together, our results show how a single HGT event followed by substantial gene duplication and neofunctionalization led to alginate catabolism and access to a new ecological niche.

## Introduction

How organisms obtain energy is a fundamental determinant of their form, function, and evolutionary trajectory. Photoautotrophs obtain energy from sunlight, while heterotrophs depend on chemical energy derived from these primary producers. Transitions between trophic strategies constitute major evolutionary steps that can lead to adaptive radiation. A prime example is the acquisition of photosynthesis through endosymbiosis between a eukaryotic heterotroph and a photosynthetic cyanobacterium that occurred around 1.5 billion years ago [1]. This key event led to the emergence of land plants, green and red algae, and glaucophytes

**Data availability statement:** All relevant data are within the paper and its Supporting information files. Raw sequencing data are available at the Sequence Read Archive under the BioProject PRJNA1205504. Additional data are available on Zenodo, including code used in the manuscript (https://zenodo.org/records/14793551) and NMR data (https://zenodo.org/records/14410447).

**Funding:** This work was supported by Temasek Life Sciences Laboratory (G.J.), the Norwegian Research Council (Project no. 226244 (Norwegian NMR Platform), 294946 (SBP-N), and 315385 (AlgModE) to F.L.A.), and the German Research Foundation (DFG, NO407/7-2 to M.N.). The funders had no role in study design, data collection and analysis, decision to publish, or preparation of the manuscript.

**Competing interests:** The authors have declared that no competing interests exist.

**Abbreviations:** A-domain, alginate lyase domain; ALY, alginate lyase; BiP, Binding immunoglobulin Protein; bp, base pair; BUSCO, Benchmarking Universal Single-Copy Orthologs; C-domain, CBM32 domain; CAZymes, carbohydrate-active enzymes; CBM32, Carbohydrate-Binding Module 32; DEH, 4-deoxy-l-erythro-5-hexoseulose urinate; DHF, 4-deoxy-d-manno-hexulofuranosidonate; ER, endoplasmic reticulum; gDNA, genomic DNA; HGT, horizontal gene transfer; OALs, oligo-alginate lyases; PL7, Polysaccharide Lyase 7; SAR, stramenopiles, alveolates, and rhizaria; SMRT, single-molecule real-time sequencing; SSW, synthetic seawater; TMD, transmembrane domain.

(Archaeplastida). Subsequent endosymbiosis between a green or red alga and eukaryotic heterotroph led to other algal lineages, which together with members of the Archaeplastida comprise the known eukaryotic photoautotrophs (reviewed in [2–5]).

While multiple endosymbiotic transitions and the diversity of eukaryotic photoautotrophs attest to the advantages of this trophic strategy, loss of photosynthesis leading to secondary heterotrophs has also occurred in each major photosynthetic lineage. In many cases, this involves a transition to parasitism and narrowing of the habitat range. Such transitions have occurred in flowering plants [6,7], green [8] and red [9] algae, and apicomplexans such as *Plasmodium* and *Toxoplasma* [10,11]. Loss of photosynthesis has also led to many free-living secondary heterotrophs. This has occurred in the green [12–15] and red [16] algae, crypto-phytes [17,18], euglenids [19], dinoflagellates [20,21], colpodellids [22], chrysophytes [23,24], and diatoms [25,26]. Here, mechanisms for nutrient uptake include phagotrophy and osmot-rophy. However, in most cases, the genetic and physiological basis for adaptation to obligate heterotrophy is poorly understood.

In the microalgal diatoms [27–29], loss of photosynthesis in the genus *Nitzschia* led to free-living heterotrophs that occupy the nutrient-rich waters of the intertidal zone. These colorless or apochlorotic diatoms have been isolated from decaying plant material and the surface of green, red, and brown algae [25,26,30–33]. Moreover, they have been shown to grow on sole carbon sources consisting of cellulose [34] and the algal cell wall polysac-charides carrageenan, agarose [31,33,34], and alginate [33]. Genome sequences have also revealed genetic signatures of apochlorotic diatoms: a β-ketoadipate pathway for metabolism of lignin-derived aromatic compounds and rewiring of mitochondrial glycolysis have been implicated in *Nitzschia* Nitz4 [35], while expansion and diversification of solute transporters, carbohydrate-active enzymes (CAZymes), and a unique secretome have been documented in *Nitzschia putrida* [36].

Here, we sequence the genome of the apochlorotic diatom *N.* sing1 and show that its ances-tor acquired a Polysaccharide Lyase 7 (PL7) family alginate lyase (ALY) gene by horizontal gene transfer (HGT) from a marine bacterium. This founder gene went on to expand through a combination of tandem gene duplication and transposition. Subsequent diversification of the paralogs gave rise to three major families comprising 91 genes. One *N.* sing1 ALY family retains the original endolytic function. By contrast, the two others underwent domain dupli-cation, gain, loss, mutation, and rearrangement to encode new functions that can account for alginate oligosaccharide import and conversion into monomers. Thus, a full alginate catabolic pathway appears to have originated through neofunctionalization of paralogs derived from a single gene obtained by HGT. ALY genes are absent from the apochlorotic diatoms *N. putrida* and *N.* Nitz4, suggesting a high degree of ecophysiological diversity within the apochlorotic lineage. Together, our data show how HGT, coupled with gene duplication and neofunction-alization, led to the evolution of a complex metabolic capability supporting the transition to obligate heterotrophy.

## Results

### *N.* sing1 obtained an alginate lyase enzyme by horizontal gene transfer

To investigate the genetic basis for obligate heterotrophy in diatoms, we sequenced, assem-bled, and annotated the genome of *Nitzschia* sing1 (*N.* sing1) (see Materials and methods). The assembly spans 40.35 Mbp and is predicted to encode 15,542 protein-coding genes (S1 Table). Analysis of genome heterozygosity indicates that vegetative *N.* sing1 cells are diploid (S1 Fig), as are other diatoms [36–41]. Putative telomeric repeats were found at both ends of one contig and at one end of another 22 contigs (S2 Table), making it likely that the *N.* sing1

genome consists of at least 12 chromosomes. Benchmarking Universal Single-Copy Orthologs (BUSCO) analysis shows that the assembly is relatively complete and comparable to that of other sequenced diatoms (S1 Table).

Because apochlorotic diatoms grow on diverse seaweed-derived polysaccharides [30,31,33,34], we searched the *N*. sing1 genome for CAZymes [42]. Here, we compared the CAZyme profile of *N*. sing1 to the two sequenced apochlorotic species (*N*. Nitz4 [35] and *N*. *putrida* [36]), five photosynthetic diatoms [37–41], and five other species in the stramenopiles, alveolates, and rhizaria (SAR) supergroup [43–49]. These data show that *N*. sing1 encodes an unusually large number (91) of genes containing at least one PL7 family ALY domain (Figs 1A and 1B and S2 and S3 Table). PL7 family ALYs are well-characterized enzymes that cleave the glycosidic bonds between alginate sugar residues through a β-elimination mechanism [50–52]. Besides the PL7 catalytic domain, the majority (77/91) of *N*. sing1 ALY genes also encode Carbohydrate-Binding Module 32 (CBM32) domains (Fig 1B). These have been shown to bind various carbohydrate moieties [53,54] and are implicated as modifiers of ALY specificity [55–57]. Interestingly, neither the photosynthetic nor other apochlorotic diatoms encode any ALY genes (Figs 1A and S2), suggesting that the acquisition and expansion of this gene family is unique to the *N*. sing1 lineage.

The *N*. sing1 ALY genes can be grouped into three families based on domain organization (Fig 1B). The most abundant group (CA family) encodes an N-terminal CBM32 domain (C-domain) followed by a single PL7 alginate lyase domain (A-domain). The two other families have A- and/or C-domains arranged in various tandem repeats. Members of the $A^{n\text{-}TMD}$ family contain no C-domains, but instead encode between 3 and 8 A-domains and a predicted C-terminal transmembrane domain (TMD). By contrast, members of the $A^n C^n$ family encode varying numbers of A- and C-domains arranged in repeats, with most encoding tandem AC repeats (Fig 1B).

*N*. sing1 occurs as an epiphyte on brown seaweeds (*Fucus serratus* [34] and *Sargassum sp*. [33]) and can catabolize alginate [33]. Thus, we focused on ALY genes as potential keys to understanding the basis for *N*. sing1's adaptive habitat invasion. To investigate the origin of *N*. sing1 ALY genes, we performed protein sequence similarity searches (BLASTP) against the non-redundant protein database (nr) using the predicted *N*. sing1 ALY proteins as query sequences. All 91 *N*. sing1 ALYs returned bacterial PL7 family members from marine bacteria as top hits, with the top-scoring bacterial ALYs also encoding N-terminal CBM32 domains. Phylogenetic analysis shows that *N*. sing1 CA family ALYs form a monophyletic group with a subset of putative ALYs encoded by marine bacteria from the genus *Vibrio* (Fig 1C). These findings strongly suggest that *N*. sing1 obtained an ancestral CBM32-containing ALY gene by HGT from a *Vibrio* marine bacterium. The majority of *N*. sing1 ALYs encode predicted signal peptides, as do the nearest relative bacterial ALYs (Fig 1B), suggesting that they are secreted. Bacterial secY- and eukaryotic Sec61-based secretory machineries have an ancestral relationship and act on hydrophobic signal peptides [58]. To assess the likely behavior of a bacterial ALY in the eukaryotic cellular environment, we fused the signal peptide encoded by a *Vibrio casei* ALY to sfGFP and expressed the fusion protein in mammalian HeLa cells. This protein is efficiently targeted to the endoplasmic reticulum (ER), presenting a fluorescent signal indistinguishable from that produced by the signal peptide from a *bona fide* ER-resident protein, Binding immunoglobulin Protein (BiP) (Fig 1D). Thus, the ALY obtained by HGT from a marine bacterium is likely to have been immediately available for secretion by the ancestor of *N*. sing1.

## Evolution of distinct alginate lyase gene families

To better understand their origin, we next constructed a phylogenetic tree from the *N*. sing1 ALY genes. To avoid biases that arise due to varying numbers of A-domains and the presence/absence of C-domains, we constructed the tree from the nucleotide sequences of each

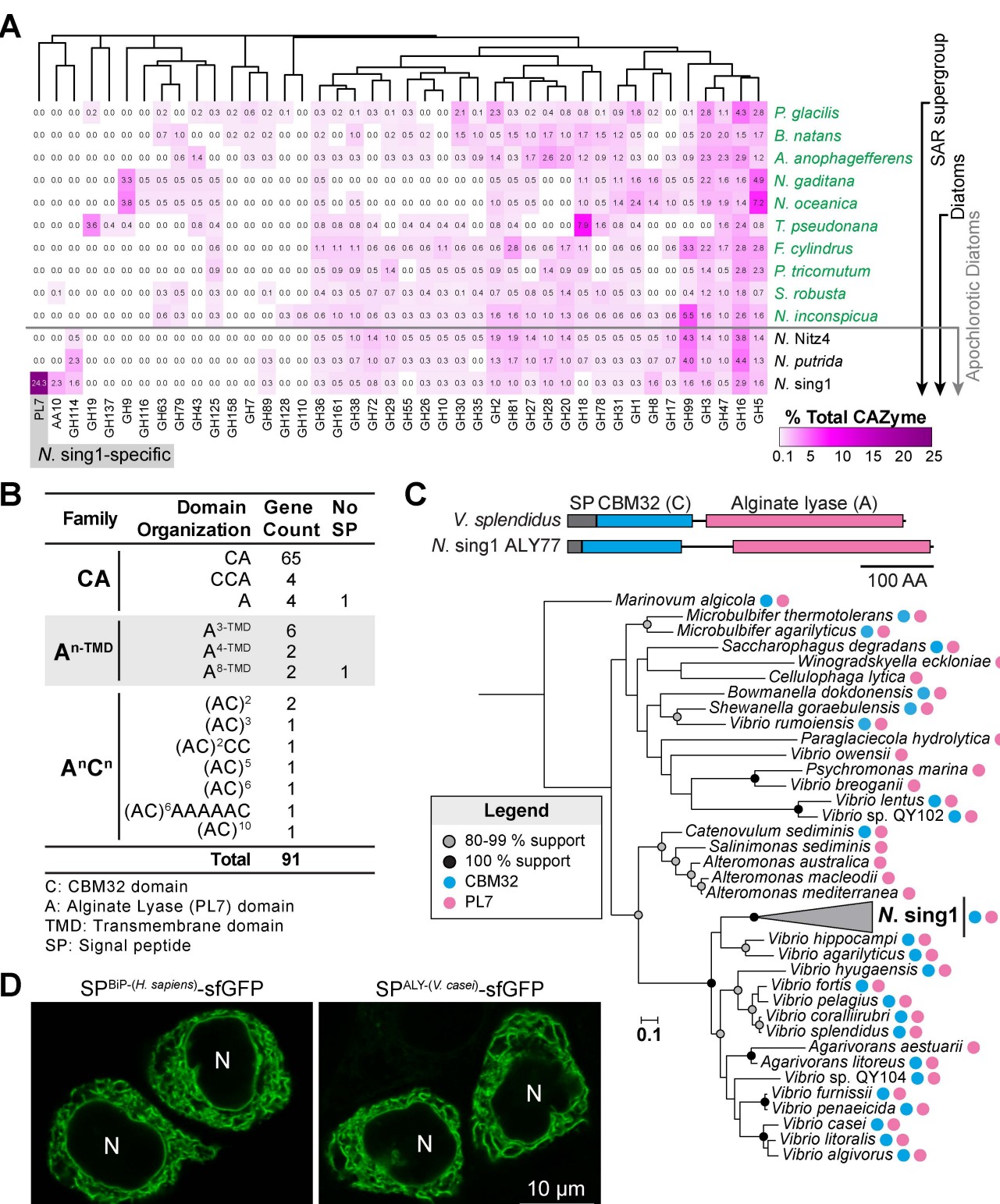

**Fig 1. *Nitzschia* sing1 acquired an alginate lyase (ALY) gene by horizontal gene transfer (HGT).** **(A)** Profile of selected carbohydrate-active enzymes (CAZymes) encoded by select diatoms and marine alga species (SAR supergroup). The heatmap displays the proportion of CAZyme family genes as a percentage of the total CAZyme genes annotated in each species (magenta scale). Polysaccharide Lyase 7 (PL7) family ALY genes only occur in *N.* sing1 (gray shade). Related to S2 Fig. The data underlying this figure can be found in S2 Data. **(B)** Ninety-one *N.* sing1 ALY genes are grouped into three families (CA, $A^{n\text{-}TMD}$ and $A^n C^n$) based on overall domain organization (N- to C-terminus). The n superscript denotes the number

of PL7 alginate lyase (A) or CBM32 (C) domains. TMD denotes transmembrane domain. The No SP column identifies the number of genes that do not encode a predicted signal peptide. The data underlying this figure can be found in S3 Data. **(C)** The cartoon illustrates the domain architecture of alginate lyases from *N. sing1* (ALY77) and *Vibrio splendidus*. The maximum likelihood phylogenetic tree (bootstrap replicates = 1,000) shows the relationship between *N. sing1* CA family ALYs and related bacteria alginate lyases. Bootstrap support for nodes and the presence of A- and C-domains are shown according to the legend. *N. sing1* ALYs are collapsed. The data underlying this figure can be found in S4 Data. **(D)** The signal peptide from a *V. casei* alginate lyase directs a GFP fusion protein (SP$^{ALY-(V. casei)}$-sfGFP) into the endoplasmic reticulum (ER) of HeLa cells. The signal peptide from the human ER luminal protein BiP (SP$^{BiP-(H. sapiens)}$-sfGFP) serves as a positive control. **N**, nucleus. Scale bar = 10 μm.

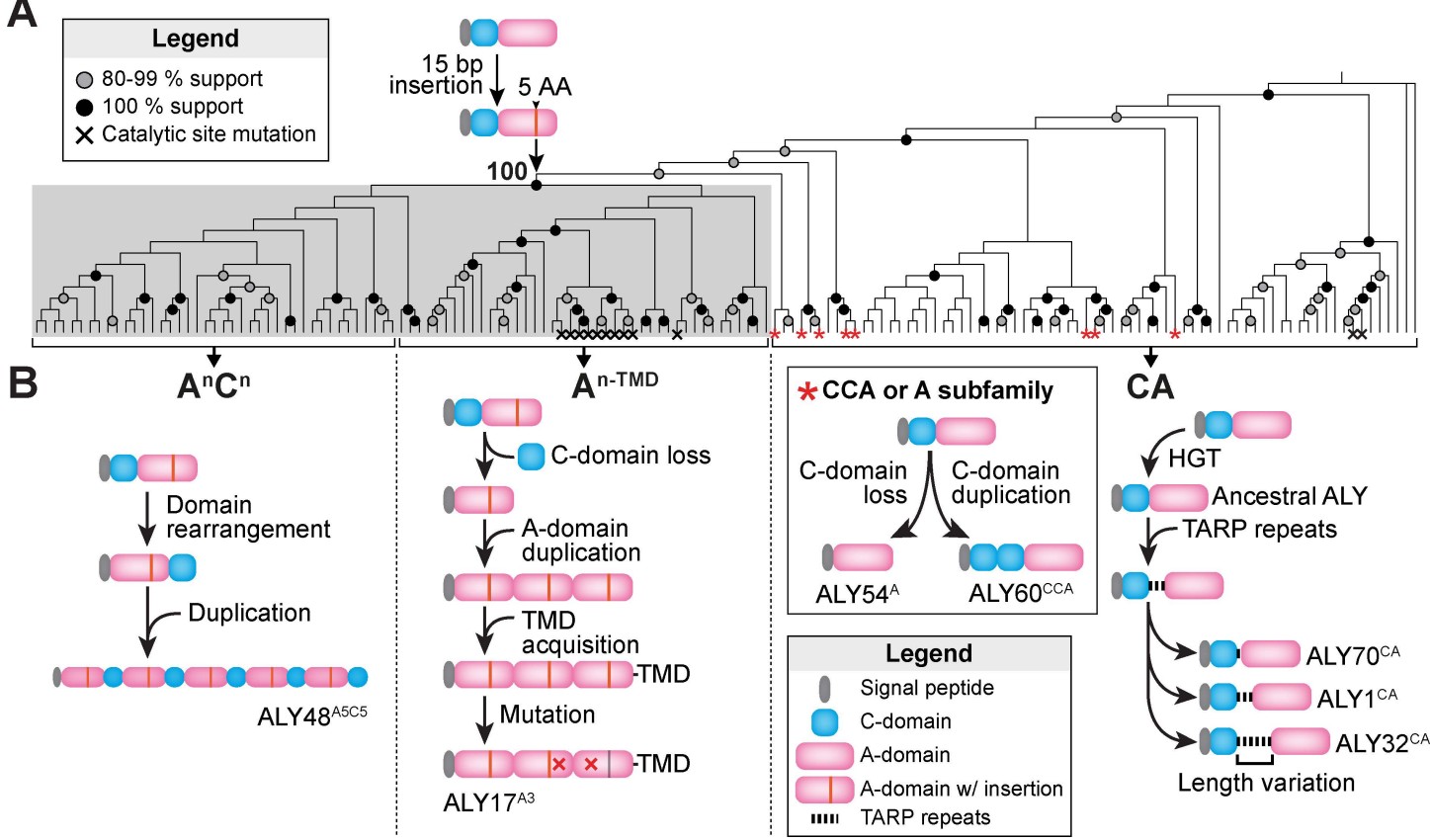

**Fig 2. Phylogeny, evolution and domain rearrangements of *N. sing1* ALYs. (A)** Maximum likelihood phylogenetic tree constructed from the nucleotide sequences of all *N. sing1* ALY A-domains (bootstrap replicates = 1,000). A-domains from the A$^{n}$-TMD and A$^{n}$C$^{n}$ family genes form a monophyletic group (shaded in gray) with 100% bootstrap support. These two families also share a 15 bp insertion, suggesting they derived from a common ancestor. A-domains with mutations in conserved catalytic residues are marked with a black cross. Domains belonging to the CCA or A subfamily ALYs are marked with a red asterisk. Bootstrap support for nodes is summarized in the legend. Related to S3 and S10 Figs. The data underlying this figure can be found in S5 Data. **(B)** Schematic diagrams illustrating the evolutionary genetic events leading to the different ALY families. Domain representations are described in the legend. Note that the order of events is arbitrary. Specific ALYs are identified for illustrative purposes.

A-domain (Figs 2A and S3). Here, genes with multiple A-domains produce multiple leaves in the tree, which are identified numerically by their position from N- to C-terminus (S3 Fig). In this tree, A-domain sequences from the A$^{n}$-TMD and A$^{n}$C$^{n}$ families form a monophyletic clade with 100% bootstrap support. Moreover, each A-domain in these two families shares a 15-base pair (bp) coding sequence insertion at precisely the same position. Thus, these two families likely originated from a common CA ancestral gene that had obtained this insertion (Fig 2A). Based on these findings, we can further infer other mutational events leading to the

two derived families—these include domain loss ($A^{n\text{-TMD}}$ family), duplication ($A^{n\text{-TMD}}$ and $A^nC^n$ families) and acquisition (TMDs of $A^{n\text{-TMD}}$ family) (Fig 2B). A small number of ALY genes lost or duplicated the C domain. These occur throughout the CA phylogeny, suggesting that they arose independently multiple times (Figs 2 and S3) and represent a CA gene subfamily.

To explore the mechanisms underlying ALY gene duplication and diversification, we next examined the genomic distribution and organization of ALY genes (Fig 3A). The 91 ALY genes are distributed across 30 loci, the majority of which (25) are found on contigs with lengths greater than 300 kb (Fig 3A), indicating that assembly fragmentation is unlikely to account for the high number of ALY loci. Twenty loci possess two or more genes arranged in tandem repeats, and all the tandem loci are supported by long reads, suggesting that the repeats are unlikely to be assembly artifacts related to repetitive DNA sequences (S4 Table). In most cases (15/20), ALY genes at tandem loci occur in a unidirectional head-to-tail (5′ to 3′) orientation with no intervening genes (Fig 3B). Such tandem duplications can be produced by homologous recombination during mitosis or meiosis [59] when recombination at misaligned repetitive DNA sequences results in one DNA strand containing a duplication and the other bearing a deletion (Fig 3C). When such unequal crossing over occurs at genes in tandem repeats—as appears to be the case with many ALY loci—intergenic regions are also duplicated. However, while coding regions can be retained through positive selection, these intergenic regions are expected to degenerate over time due to genetic drift. We next used nucleotide sequence dot plots to search for patterns of homology at duplicated ALY loci (S4–S8 Figs). We further developed a method to quantify these dot plots and graphically show levels of intergenic homology as matrix plots (Figs 3C and S9). As expected, homologies are detected between all ALY coding regions at any given locus. In addition, some loci also display high levels of homology between consecutive intergenic regions (see loci 7, 8, 9, 19, 25, 26, 28, and 29) (Figs 3C and S9). Thus, these are likely to constitute the most recent duplication events. These data further provide definitive evidence for ALY gene duplication through unequal crossing over.

Data presented thus far point to acquisition of a CA family ALY gene by HGT followed by ALY gene duplication through unequal crossing over. However, ALY loci are dispersed across the genome in a manner that is likely due to transposition events [60]. To explore this mechanism, we examined the phylogenetic relationships between genes encoded at the various ALY loci (Fig 3B) and assigned a possible relationship through transposition when genes at two or more unlinked loci (separated by unique sequences greater than 10 kb in length) form monophyletic clades with substantial statistical support (100% bootstrap support). Employing a stringent cutoff identifies such events with confidence, but is also likely to exclude transposition-derived loci where greater sequence divergence has occurred between genes. Relationships through putative transposition events are almost exclusively found within individual ALY families, indicating that the three families probably evolved unique identities prior to duplication through transposition (Fig 3B). Only one locus contains genes from different families. Here, locus 25 consists of seven CA genes and one $A^nC^n$ gene ($ALY67^{A2C2}$). Because the latter shows a close phylogenetic relationship to $ALY82^{A2C2}$ encoded at locus 27, we conclude that the two are related through transposition. Together, the data document an initial ALY gene acquisition by HGT, diversification into three families through domain rearrangement and mutation, and ALY gene family expansion by tandem duplication and transposition.

## Enzymatic activity of *N.* sing1 alginate lyases

Alginate comprises up to 40% of brown algal biomass [61] and consists of linear chains of β-D-mannuronate (M) and its C5 epimer α-L-guluronate (G) linked by 1,4 glycosidic bonds. M and G residues can occur as Poly-M, Poly-G, and alternating Poly-MG-enriched blocks [62],

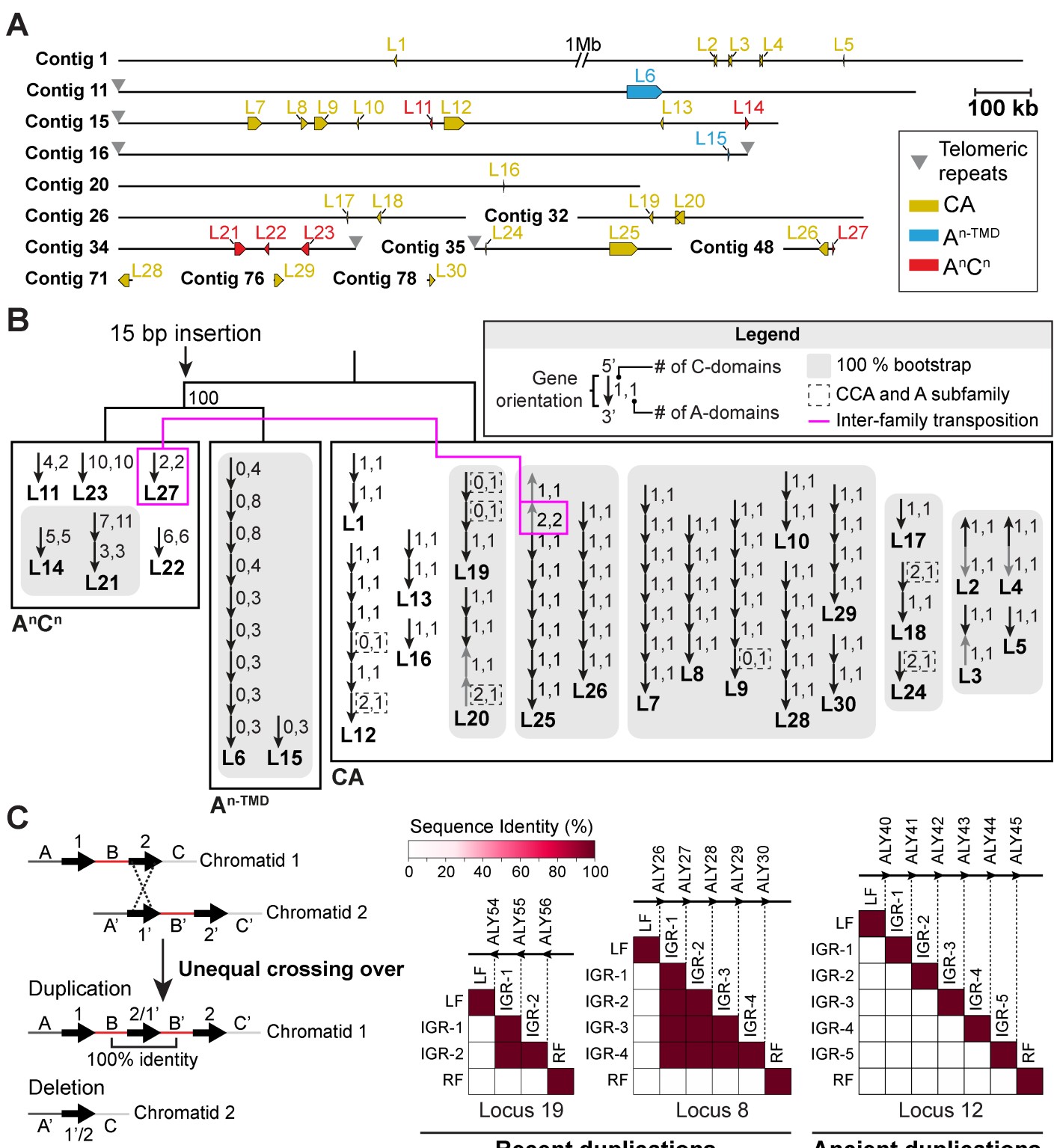

**Fig 3. ALY gene duplication through unequal crossing over and transposition. (A)** Locations of ALY loci on the *N.* sing1 contig assembly. ALY loci (L) and telomeric repeats are identified according to the legend. **(B)** A simplified version of the phylogenetic tree from Fig 2A (midpoint rooted) is shown along with the arrangement of ALY genes at all 30 loci. Each arrow represents a single ALY gene, which is annotated according to the legend. Genes oriented opposite to the rest of the locus are colored in gray. Loci where most genes form a monophyletic clade with 100% bootstrap support are shaded in gray and are presumed to be related through transposition. The magenta line identifies a unique case where transposition appears to have produced a locus (L25) encoding ALYs from two different

families. **(C)** Homology between intergenic sequences at ALY loci identifies recent tandem duplication events. The cartoon illustrates how unequal crossing over can lead to duplication of the gene and intergenic region. The matrix plots show self-comparisons (percent sequence identity) of intergenic sequences at the indicated loci. Recent duplication events are inferred through high levels of homology between consecutive intergenic pairs (loci 8 and 19). By contrast, presumably ancient duplications (loci 12) show no intergenic sequence homology. IGR, intergenic region; LF, left flank; RF, right flank. Related to S4–S9 Figs. The data underlying this figure can be found in S6 Data.

which influences the physicochemical properties of the polymer [63]. Notably, Poly-G regions self-associate through divalent cation binding [64] to form hydrogels, which play a structural role in the cell wall [65,66]. Marine bacteria generally initiate alginate breakdown by secreting endolytic PL7 family ALYs, which convert alginate polymers into short oligosaccharides. These are imported into the cell through various transporters before being converted into monomers by exolytic oligo-alginate lyases (OALs). Alginate monomers then enter central carbon metabolism to yield ATP and pyruvate [67]. Similarity searches failed to return any clear *N. sing1* homologs of bacterial transporters or OAL enzymes (S5 Table), suggesting that *N. sing1* executes these steps through distinct machineries.

We next examined the enzymatic activity of ALY A-domains to determine how they contribute to alginate catabolism. PL7 ALYs possess well-defined catalytic residues [50,68] that are largely conserved within the CA and $A^nC^n$ families (Fig 2A). However, within the $A^{n-TMD}$ family, many A-domains have substitutions at key catalytic residues (Figs 2A and S10). To examine ALY enzyme activity, we developed an assay where alginate gel liquefaction—which occurs with alginate degradation—is measured by fluid displacement upon vortexing (see Materials and methods). A-domain representatives from each family were expressed in *Escherichia coli* and crude extracts were examined for the ability to promote liquefaction. Despite some being found exclusively in inclusion bodies, all six CA family ALYs are active and display similar liquefying activities as compared to a positive control consisting of a commercially available endolytic ALY. By contrast, A-domains from both $A^{n-TMD}$ and $A^nC^n$ family members are inactive or display very low activity (Fig 4A).

We next directly measured endolytic enzyme activity through UV absorption by double bonds that form at non-reducing ends of oligosaccharide products. Attempts to purify soluble A-domains were met with limited success due to inclusion body formation. However, we were able to produce ALY7-A, ALY58-A, ALY77-A (CA family), and ALY48$^{A5C5}$-A1 ($A^nC^n$ family) as soluble enzymes. Here, in keeping with the liquefaction assay, all CA family A-domains are enzymatically active, while the $A^nC^n$ family A-domain appears to be inactive (Fig 4B).

To investigate how the 15-bp insertion in the A-domain of ALY48$^{A5C5}$-A1 might impact its enzyme activity, we used AlphaFold2 [69] to predict its structure along with the active endolytic CA enzyme, ALY77-A. The predicted structure of both proteins closely resembles the β-jelly roll fold seen in PL7 ALY crystal structures [50] (Figs 4C and S11). Interestingly, in ALY48$^{A5C5}$-A1, the 5 amino acids encoded by the 15-bp insertion forms a surface loop that appears to occlude the alginate-binding groove to form a binding pocket. This could promote alginate end-binding and convert the enzyme from an endolytic to exolytic mode of action. A similar structural change has previously been correlated with the emergence of exolytic activity in a PL7 family alginate lyase (AlyA5) from the bacterium *Zobellia galactanivorans* [70].

To determine the specificity of endolytic CA family A-domains and the possibility that ALY48$^{A5C5}$-A1 is an exolytic ALY, we next employed NMR to directly observe reaction products [71] (Figs 4D and 4E and S12–S15). These data confirm that all three CA family A-domains are endolytic enzymes that accumulate oligosaccharides between two and six residues in length. In addition, they are inactive on Poly-M and generally show distinct preferences for cleavage at G|GG and G|MG sites. None of these enzymes produced monomers, indicating that they are

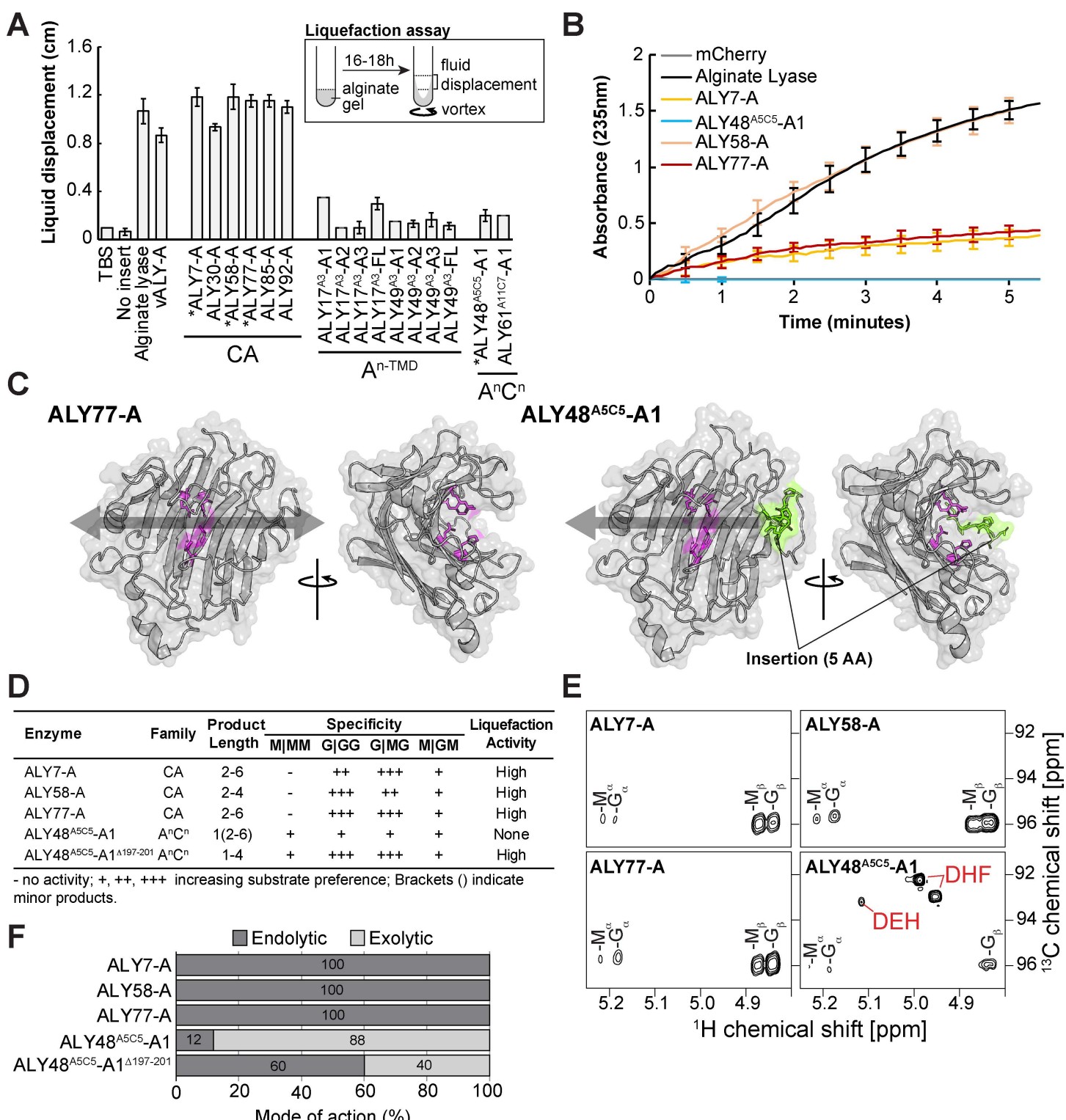

**Fig 4. N. sing1 ALY enzyme activity and substrate specificity. (A)** Alginate gel liquefaction assay using lysates from *Escherichia coli* expressing the indicated proteins. Liquefaction is measured as shown in the legend. Buffer alone (TBS) and a lysate from *E. coli* expressing an empty vector (No insert) serve as negative controls. Commercial endolytic alginate lyase (Alginate lyase) serves as a positive control. Asterisks identify the proteins that could be purified in a soluble form. The data underlying this figure can be found in S7 Data. **(B)** Alginate lyase activity of the indicated recombinant proteins as measured by an increase in absorbance at 235 nm. Standard deviation is shown. mCherry serves as a negative control while a commercial endolytic alginate lyase (Alginate lyase) serves as a positive control. The data underlying this figure can be found in S7 Data. **(C)** AlphaFold2 predictions of A-domain structure from representative CA (ALY77-A) and AⁿCⁿ (ALY48^{A5C5}-A1) family proteins.

Catalytic side chains are colored in magenta. Amino acid side chains encoded by the 15 bp insertion are colored in green. The arrows show how an alginate polymer might bind to the enzymes. Related to S11 Fig. The data underlying this figure can be found at https://zenodo.org/records/14793551. **(D)** The table summarizes the enzymatic activity of the indicated A-domains with product length, specificity and liquefaction activity indicated. **(E)** Alginate reaction products produced by the indicated A-domains. The panels show a region of the NMR spectrum ($^1$H-$^{13}$C HSQC) where alginate monomers DEH (4-Deoxy-L-erythro-5-hexoseulose uronate) and DHF (two epimers of 4-deoxy-D-manno-hexulofuranosidonate) occur (labeled in red). Related to S12–S15 Figs. The data underlying this figure can be found at https://zenodo.org/records/14410447. **(F)** ALY enzyme mode of action. Endolytic and exolytic mode of action for the indicated enzymes is estimated through the integration of NMR peaks (see Materials and methods). The data underlying this figure can be found at https://zenodo.org/records/14410447.

exclusively endolytic. By contrast, the ALY48$^{A5C5}$-A1 domain displays an exolytic activity on all alginate substrates and accumulates the monomer products 4-Deoxy-L-erythro-5-hexoseulose urinate (DEH) and 4-deoxy-D-manno-hexulofuranosidonate (DHF) (Fig 4E). Together, these findings indicate that CA family members have retained the ancestral endolytic function, while the A$^n$C$^n$ A-domain acquired an exolytic mode of action (Figs 4D and 4E, and S12–S15).

Finally, to directly assess the influence of the 15-bp insertion on the ALY48$^{A5C5}$-A1 enzyme, we removed the insertion to generate the mutant ALY48$^{A5C5}$-A1$^{\Delta197-201}$. The preferred mode of enzyme action (endolytic or exolytic) was then estimated by integrating reaction product NMR signals of monomer and oligosaccharide products for each enzyme (Fig 4F). This data shows that ALY48$^{A5C5}$-A1$^{\Delta197-201}$ has a substantially increased endolytic activity as opposed to wild-type ALY48$^{A5C5}$-A1. This activity is further corroborated by high activity in the liquefaction assay for the mutant enzyme (Fig 4D). Together, these findings confirm the role of the insertional mutation in switching the A$^n$C$^n$ family enzymes from an endolytic to exolytic mode of action.

## A role for vacuoles and endomembrane trafficking in alginate uptake

To better understand the cell biology of *N.* sing1 alginate catabolism, we imaged *N.* sing1 cells grown in seawater medium supplemented with alginate and dextrose as sole carbon sources. Alginate forms hydrogels in seawater that can be seen as amorphous aggregates with brightfield microscopy (Fig 5A). Over time, *N.* sing1 diatoms begin to degrade alginate, as evidenced by the increase in absorbance at 235 nm. The medium also turns yellow and develops a broad absorbance shoulder that tails off at around 500 nm (Fig 5B). In addition, alginate hydrogels also start fluorescing when viewed under epifluorescence (Fig 5A). Interestingly, vacuoles of diatoms grown on alginate display a related fluorescence, while those grown on dextrose do not (Fig 5C). The emission spectra of these vacuoles and the alginate seawater medium from a saturated *N.* sing1 culture overlap (Fig 5E), suggesting that the vacuole-derived fluorescence is indeed due to uptake of alginate-related fluorescent moieties. To further examine this phenomenon, we employed the vacuole dye CMAC (7-amino-4-chloromethylcoumarin) [72] and found that vacuoles in alginate-grown cells are substantially larger and more abundant than those observed in dextrose-grown cells (Figs 5C and 5D and S16A). Together, these data suggest that products of alginate breakdown are taken up by endocytosis and processed within vacuoles. When *Vibrio* sp. bacteria are grown on the same alginate medium, absorbance at 235 nm is detected. However, no additional absorbance (Fig 5B) nor medium yellowing is observed (S16B Fig). Thus, these aspects of alginate metabolism appear to be *N.* sing1-specific.

A$^{n-TMD}$ family members possess mutations in catalytic domains (S10 Fig) and appear to be enzymatically inactive (Fig 4A). These all encode a predicted N-terminal signal peptide and a family-specific C-terminal TMD. This configuration is expected to result in cell surface proteins anchored to the plasma membrane with A-domains projecting into the extracellular milieu. Based on this, we hypothesize that they function as alginate import receptors. As attempts to transform *N.* sing1 to express an A$^{n-TMD}$ fluorescent fusion protein were unsuccessful, we turned to heterologous expression in a mammalian cell line. A full-length

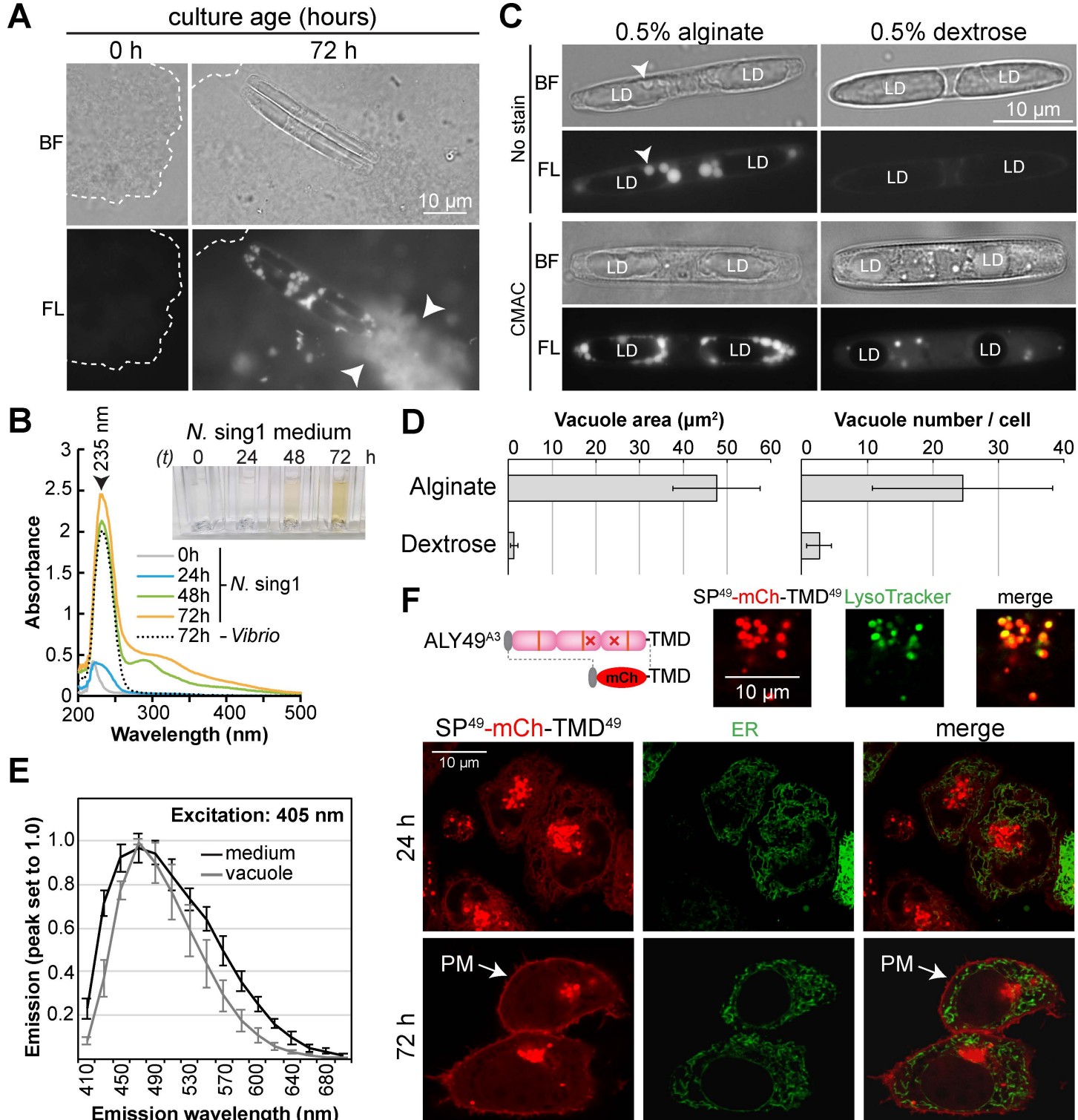

**Fig 5. Cell biology of *N. sing1* alginate metabolism. (A)** Alginate hydrogels in seawater medium are seen as amorphous aggregates by brightfield microscopy (BF, outlined). Note that they begin to fluoresce as *N. sing1* diatoms grow (FL, 72 h, arrowheads). Scale bar = 10 μm. **(B)** Absorbance spectra of alginate culture medium from *N. sing1* and *Vibrio* sp. cultures at different time points. The absorbance peak at 235 nm corresponds to non-reducing ends produced by alginate lyase activity. Note the absorbance shoulder (270–450 nm) that occurs with *N. sing1*, but not *Vibrio* sp. cultures. The inset shows the appearance of diatom culture media at various time points. Related to S16B Fig. The data underlying this figure can be found in S9 Data. **(C)** Vacuoles in *N. sing1* (arrowheads) fluoresce in cells grown on alginate, but not dextrose (LD: lipid droplets). CMAC staining reveals the proliferation of vacuoles in alginate as compared to dextrose cultures. Scale bar = 10 μm. **(D)** Quantification

of vacuole size and number in cells grown on alginate or dextrose ($n$ = 10). Related to S16A Fig. The data underlying this figure can be found in S9 Data. **(E)** Emission spectra (excitation: 405 nm) of medium (black) and *N*. sing1 vacuoles (gray) after 72 h growth. The data underlying this figure can be found in S9 Data. **(F)** Localization of mCherry fusion protein with the signal peptide (SP) and transmembrane domain (TMD) from ALY49[A3] (SP[49]-mCh-TMD[49]) in HeLa cells. The perinuclear organelles are defined as lysosomes by colocalization with LysoTracker (upper panels). sfGFP fused with a signal peptide from the human endoplasmic reticulum (ER) protein BiP and the ER retention signal (KDEL) serves as an ER marker (ER). Note that SP[49]-mCh-TMD[49] localization shifts from the ER and lysosomes at 24 h to the plasma membrane (PM) and lysosomes at 72 h. Scale bar = 10 μm.

mCherry-ALY49[A3] fusion protein did not appreciably accumulate. However, we were able to visualize an mCherry fusion protein encoding the N-terminal signal peptide and C-terminal TMD from ALY49[A3] (Fig 5F). At early time points after transfection (24 h), the fusion protein is detected in both the ER and lysosome (colocalization with the ER marker BiP and LysoTracker, respectively). This indicates a signal peptide-dependent insertion into the ER, followed by endomembrane trafficking to lysosomes. At later time points (72 h), the fusion protein localizes to the cell surface and lysosome, but not in the ER (as expected of transient transfections). This pattern of steady-state localization suggests cycling between the plasma membrane and lysosome. Overall, this trafficking pattern is consistent with the proposed alginate import function.

## Disordered repetitive domains evolved in the *N. sing1* lineage and bind to alginate

Data presented thus far document the evolution of ALY protein novelties in both enzymatic activity (Fig 4) and protein trafficking (Fig 5F). We next examined low-complexity sequences that occur between the two domains of CA family proteins (Fig 2B). These are predicted to be disordered (S17A Fig) and are enriched for tetrapeptide sequences with a T-(A/P/S/V)-R-P consensus motif ("TARP" repeats). Importantly, these repeats are not observed in bacterial ALYs, indicating that they evolved independently in the *N*. sing1 lineage. Protein sequence alignments show that the length of these TARP-containing regions tends to vary precisely in the number of TARP repeats (Figs 6A and S17B). These regions also have a net positive charge through arginine that could bind to negatively charged alginate. To explore this idea, we produced mCherry fusion proteins to a diverse group of TARP repeat regions (S18A Fig), along with a C-domain from the CA family gene ALY77 (ALY77-C) fused to either mCherry or mGFP. ALY77-C-mGFP binds to calcium alginate hydrogels as evidenced by fluorescence imaging (Figs 6B and S18B), a pelleting assay (S19A Fig) and FRAP (S19B Fig). TARP repeat regions also bind to alginate gels; however, they produce fluorescent puncta distinct from the uniform binding pattern observed with the C-domain (Figs 6B and S18B). In general, naturally occurring TARP sequences with more repeats produce brighter fluorescent puncta (S18B Fig). To exclude the influence of other sequences found in these regions, we next produced mCherry fusions with varying lengths of consecutive TARP repeats (3, 6, 9, and 12). Fluorescence microscopy shows that alginate binding by these sequences is indeed cooperative, with those containing more TARP repeats producing brighter fluorescence signal (Fig 6B).

To investigate the role of arginine in TARP repeats, we chose the ALY1 TARP domain (TARP1), which contains nine TARP repeats, and produced variants that substitute arginine with either serine or lysine. Substitutions with serine (R→S) abolish alginate binding, while substitutions with positively charged lysine (R→K) diminish binding (Fig 6B). These data indicate that electrostatic interactions promote alginate binding by TARP repeats. To investigate whether TARP and C-domains also bind to uncomplexed alginate, we incubated the recombinant fusion proteins with alginate in the absence of calcium. Fluorescence microscopy shows that TARP proteins form complexes with soluble alginate to produce a fine punctate

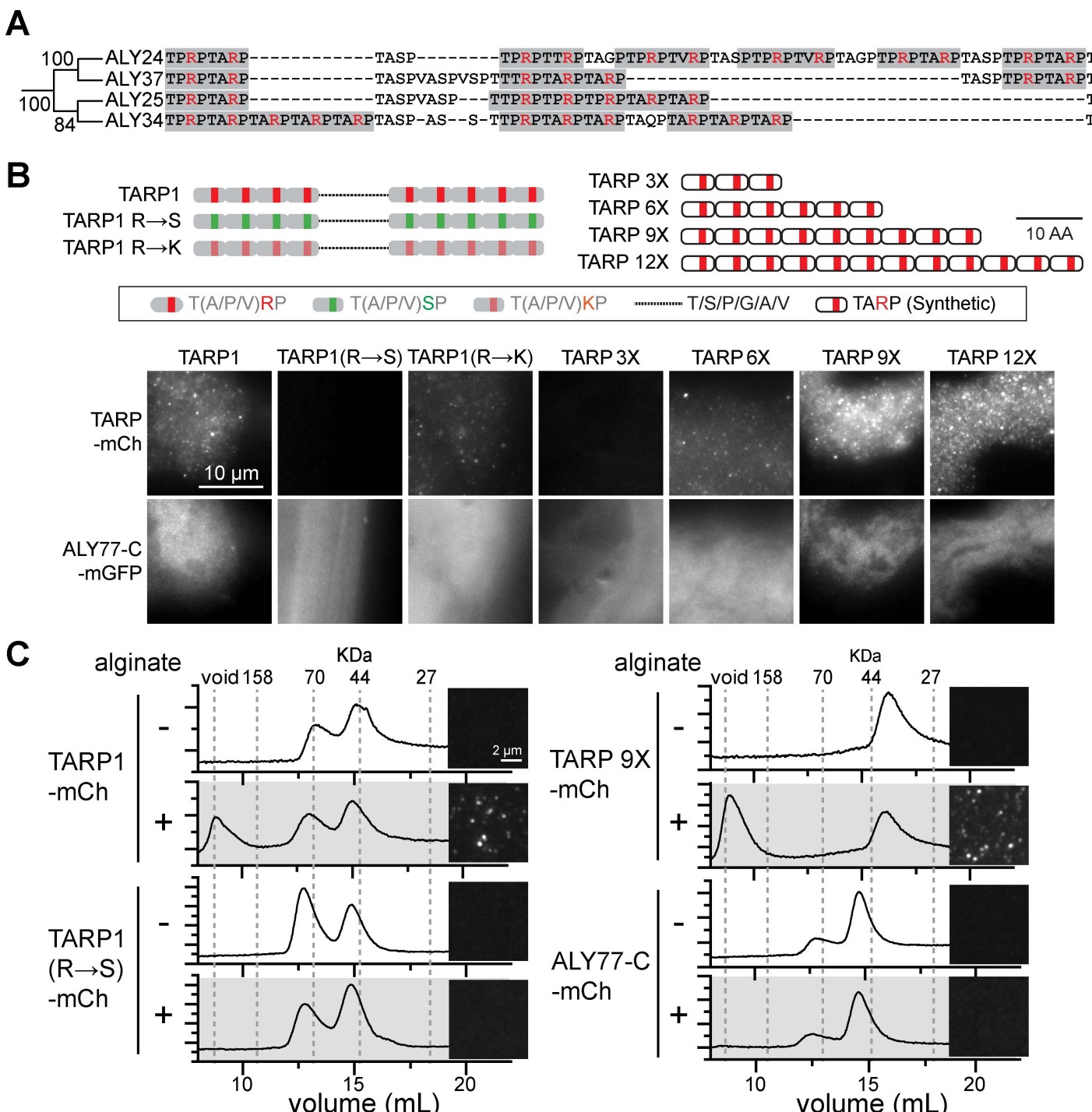

**Fig 6. TARP repeat sequences bind to alginate.** (A) Closely related CA proteins show variation in TARP repeat length. TARP repeats are highlighted in gray, and arginine (R) residues are colored red. Related to S17B Fig. The data underlying this figure can be found in S10 Data. (B) The cartoons depict TARP repeat variants. TARP1 is the naturally occurring sequence found between the C- and A-domains of ALY1. R→S and R→K denote TARP1 with arginine substitutions to serine and lysine, respectively. TARP 3/6/9/12X are synthetic TARP repeat length variants. TARP repeat-mCherry fusion proteins bind calcium alginate hydrogels to produce a punctate pattern of fluorescence distinct from the uniform signal produced by the C-domain from ALY77 (ALY77-C-mGFP). R→S substitution abolishes binding while R→K substitution diminishes binding. Length variants show that TARP repeats promote binding in a cooperative manner. Scale bar = 10 μm. Related to S18 Fig. (C) TARP repeats bind to soluble alginate. Gel filtration chromatograms of TARP variants and ALY77-C-mCherry in the presence (+) or absence (−) of alginate. When alginate is present, both TARP1-mCh and TARP 9X-mCh shift into the void volume, indicating the formation of a macromolecular complex. This

shift is not observed with ALY77-C-mCherry or the TARP1 R→S variant. Images next to each chromatogram show the samples as seen by epifluorescence. Note that fluorescent puncta are only observed in samples that display a void volume shift in the chromatogram. Scale bar = 2 μm. The data underlying this figure can be found in S11 Data.

signal. By contrast, the C-domain and TARP1 R→S variant do not produce any detectable signal (Fig 6C). These observations are corroborated by gel filtration, where a shift in the TARP1 elution peak to the void volume indicates formation of a high molecular weight complex between TARP1 and alginate. This shift is abolished with the TARP1 R→S variant and does not occur with the C-domain (Fig 6C). Together, these data show that C-domains bind to structural moieties associated with the calcium alginate hydrogel, while TARP repeats appear to bind uncomplexed alginate through arginine-based electrostatic interactions.

## Discussion

Apochlorotic diatoms lost photosynthesis to become free-living heterotrophs. Here, we account for their acquisition of alginate catabolism and adaptive invasion of brown algal habitats. The ancestor of *N.* sing1 appears to have acquired an ALY gene by HGT from a *Vibrio* marine bacterium (Fig 1). From this basis, the capacity to degrade alginate evolved through gene duplication and neofunctionalization. Unequal crossing over and transposition underlie duplication (Fig 3), while domain duplication, gain, loss, mutation, and de novo sequence evolution led to the advent of new protein activities. These include ALYs with exolytic enzyme activity (Fig 4), putative alginate import receptors (Fig 5) and low-complexity alginate-binding sequences (Fig 6). Thus, while HGT from a primary heterotroph provided the foundation for alginate catabolism, evolving the full capacity required a substantial amount of lineage-specific genetic innovation.

The ancestral endolytic function of the PL7 family ALYs is retained in *N.* sing1 CA family ALYs (Fig 4). However, alginate catabolism also requires mechanisms for the import of alginate oligosaccharides and their digestion into monomers. Marine bacterial alginate degradation systems are well-characterized [50,52,67]: In *Vibrio* sp., extracellular alginate oligosaccharides produced by endolytic ALYs [50,52] are imported through transporters [67] and subsequently converted to monomers by cytoplasmic OAL enzymes. *N.* sing1 lacks bacteria-like transporters and OALs (S5 Table). Instead, monomer production can be accounted for by the emergence of exolytic activity in the $A^nC^n$ family A-domains (Figs 4D–4F and S12–S15), while alginate import appears to depend on the endomembrane system (Fig 5C and 5D), with $A^{n\text{-TMD}}$ family members potentially acting as import receptors (Fig 5F). Alginate-binding TARP repeats represent an additional form of innovation that appears unique to the *N.* sing1 lineage.

HGT can drive eukaryotic evolution when an acquired gene promotes access to a new ecological niche or lifestyle. Such acquisitions have been linked with adaptation to environmental extremes, parasitism, and nutrient acquisition (reviewed in [73–76]). Genes acquired by HGT appear to frequently undergo duplication [77–83]. However, relatively few studies have performed functional assays to document new protein activities in duplication-derived paralogs [84–87]. In two such cases, duplication-derived paralogs evolved new activities related to nutrient acquisition [85,86], as documented here for ALY genes.

Most ALY gene tandem duplicates are arranged in a head-to-tail orientation (Fig 3B), suggesting that they expanded through unequal crossing over during homologous recombination. High levels of intergenic sequence homology at certain ALY loci provide definitive evidence for this conclusion (Figs 3C and S4–S9) and indicate that the evolution of alginate catabolism

is likely to be ongoing. Unequal crossing over can account for gene duplication as well as the domain duplications required for the emergence of $A^{n-TMD}$ and $A^nC^n$ families (Figs 2 and 3). We conclude that unequal crossing over was a major source of genetic variation underlying the evolution of alginate catabolism. It may also account for the expansion of low-complexity TARP repeat sequences. However, in this case, an origin through other mechanisms such as errors in DNA replication cannot be excluded.

Tandem duplicates commonly constitute a significant proportion (~10%) of eukaryotic gene content [88–90] and their expansion has been associated with the evolution of environmental sensing in plants [91] as well as adaptation to benthic marine habitats in diatoms [40] and shrimp [92]. To the best of our knowledge, intergenic homology has not been employed to identify recent duplication events as shown here for ALY loci (Figs 3C and S9). The systematic application of this approach has the potential to identify actively duplicating loci related to ongoing adaptive change. Interestingly, lab-based evolution experiments in the diatoms *Seminavis robusta* and *Phaeodactylum tricornutum* documented high rates of mitotic homologous recombination [93]. This suggests that duplications may frequently arise through unequal crossing over in vegetative populations. These findings raise a scenario where much of the genetic variation that led to the evolution of alginate catabolism in the *N*. sing1 lineage had an asexual origin. Lab-based evolution experiments with *N*. sing1 will help to address this possibility.

The CA family expanded to comprise 73 members and the majority of these retain signal peptides and catalytic residues (Figs 1B and 2A). Gene dosage effects and diversification of enzyme specificity towards different alginate linkages are likely to provide a selective advantage for CA gene expansion and maintenance. Interestingly, in closely related *Vibrio* strains, HGT leading to high ALY copy number is correlated with increasing enzyme activity and efficiency of alginate utilization [94]. Thus, gene dosage effects appear to have played a role in the adaptive evolution of alginate catabolism in both bacteria and eukaryotes. The C-domain was acquired from the founding bacterial ALY gene, while TARP repeats appear to have arisen in the *N*. sing1 lineage. Both bind to *Sargassum* cell walls, (S19C Fig) but interact with alginate through distinct mechanisms (Fig 6). Future work can examine whether these domains cooperate and how they influence alginate processing by A-domains.

$A^{n-TMD}$ family members acquired a C-terminal TMD (Figs 2B) and lost enzymatic activity (Fig 4A) through mutation (Figs 2 and S10). Heterologous expression (Fig 5F) indicates that $A^{n-TMD}$ family members are likely to be tethered to the cell surface to present A-domains to the extracellular space. Growth of *N*. sing1 on alginate is accompanied by the formation of alginate-related fluorescent moieties that enter vacuoles (Figs 5A and 5C). Moreover, vacuoles become enlarged and increase in numbers when diatoms are grown on alginate as compared to dextrose (Figs 5C and 5D). Thus, while the nature of fluorescent alginate-related moieties remains to be determined, endocytosis and vacuoles appear to provide for alginate internalization. $A^nC^n$ family members have signal peptides for entry into the secretory pathway (Fig 1B) and can convert alginate oligosaccharides into monomers (Figs 4E and S12). However, whether they are secreted outside the cell or undergo endomembrane transport to obtain residency in the vacuole remains to be determined. Other open questions concern how alginate-derived sugars are transported from the vacuole lumen to the cytoplasm, as well as how they enter central carbon metabolism. Additional work and the development of techniques for genetic transformation and gene editing in *N*. sing1 will help to address these questions.

Marine fungi comprise the other group of eukaryotes that are known to catabolize alginate [95]. The genome sequence and proteomic analysis of one such fungus, *Paradendryphiella salina,* identified three secreted PL7 family ALYs that are induced by the presence of alginate [96]. These data support the idea that marine fungi and apochlorotic diatoms initiate alginate

catabolism through secreted endolytic ALYs. Interestingly, another marine fungus, *Asteromyces cruciatus,* encodes a fungus-specific transporter*, Ac_DHT1,* which confers alginate monomer import upon heterologous expression in yeast [97]. *N.* sing1 does not encode Ac_DHT1 homologs (S5 Table). Thus, marine fungi and diatoms appear to be distinct in their mechanism of alginate import.

Diatoms possess features that are likely to have supported the transition to obligate heterotrophy. These include high levels of intraspecific genetic variation [39,40,98], a predisposition towards rapid evolution [38–40,93], mixotrophy [99–102], diverse metabolic networks [103–106], and substantial force generation through gliding motility [33]. *N.* sing1 and phylogenetically distinct relatives catabolize alginate, while *N. putrida* does not [33]. Moreover, *N. putrida* and *N.* Nitz4 do not encode ALY genes (Fig 1A). Thus, current data point to a high degree of ecophysiological diversity in the apochlorotic lineage. More environmental sampling and genome sequencing will be required to better understand the biodiversity and heterotrophic strategies of these diatoms. Going forward, apochlorotic diatoms will make good models to study the genetics and physiology of adaptive radiation while also expanding our understanding of nutrient cycling at the intertidal zone.

## Materials and methods

### Diatom culture conditions and nucleic acid extraction

*Nitzschia* sing1-1 diatoms were cultured on synthetic seawater (SSW) medium as previously described [33]. For genomic DNA (gDNA) and total RNA extraction, diatoms were grown in liquid SSW medium supplemented with 0.5% (w/v) dextrose (Sigma-Aldrich, D9434) at 30 °C for 3 days. For cell imaging experiments (Fig 5A–5E), diatoms were grown in SSW medium supplemented with either 0.5% (w/v) dextrose or medium-viscosity sodium alginate (Sigma-Aldrich, A2033). *N.* sing1 gDNA was extracted using the MasterPure Yeast DNA Purification Kit (Lucigen, MPY80200). Total RNA was extracted with TRIzol (Invitrogen, 15596018), treated with DNase I (Roche, 04716728001) at 37 °C for 20 min and purified by Phenol/Chloroform extraction.

### Genome sequencing and assembly

*N.* sing1 gDNA was sequenced using the Pacific Biosciences (PacBio) single-molecule real-time sequencing (SMRT) technology (Sequel II system, Continuous Long Read (CLR) mode) (NovogeneAIT Genomics) for long-reads and BGI's cPAS and DNA nanoballs technology (BGISEQ-500, DNB-seq) (BGI Tech Solutions) for short-reads. PacBio long-read sequencing yielded 7.75 Gb (608,716 subreads, mean read length = 12.5 kb) after filtering (SMRT Link parameters: minLength 0, minReadScore 0.8) and partitioning. Paired-end short-read sequencing yielded 4.24 Gb (28,322,876 reads, read length = 150 bp). Adapter and low-quality short-read sequences were filtered using the SOAPnuke [107] software (SOAPnuke parameters: -n 0.01 -l 10 -A 0.25 -Q 2 -G -minLen 150).

PacBio sequence long-reads were assembled using HGAP4 (Hierarchical Genome Assembly Process 4) [108] from SMRT Tools (SMRT Link v10.1.0 package), resulting in 80 contigs with a total length of 40.4 Mb. This initial assembly was subjected to two rounds of correction with Pilon (v1.24) [109] based on genomic BGISEQ short-reads. BLAST comparisons [110,111] against mitochondrial proteins from *Nitzschia alba* (GenBank accession: NC_037729.1) [112] revealed two contigs that most likely correspond to mitochondria DNA. These two contigs, which also have a much lower GC content (29%) than other contigs (45%), were removed from the final assembly. The final genome assembly consists of 78 contigs with a total size of 40.35 Mb (S1 Table).

To assess the quality of the genome assembly, BUSCO (v5.5.0) [113] assessments were performed using the eukaryotic (eukaryote_odb10) and stramenopiles lineage datasets (stramenopiles_odb10). Genome assembly statistics were also computed with QUAST (v5.2.0) [114] (S1 Table). *K*-mers were counted using the Jellyfish software (v2.3.1) [115] and genome heterozygosity was modeled using GenomeScope (v1.0.0) [116] (S1 Fig). Analysis of transposable and other repeat elements was performed as described in Traeger and colleagues (2013) [117]. Briefly, repeats in the *N*. sing1 genome were identified using RepeatMasker (v4.1.5; rmblastn engine v2.14.0) [118] with a de novo repeat library generated from RepeatModeler (v2.0.5) [119]. Putative telomeric repeats (sequence TTAGGG) were identified using a custom Perl script (S2 Table).

BLAST analysis on one of the two putative mitochondrial contigs showed that they contain all the proteins predicted in the *N*. *alba* mitochondrial genome [112]. These contigs also have a similar size of 37 kb (compared to 36.2 kb of *N*. *alba*), further suggesting that they represent the mitochondrial genome of *N*. sing1. Contigs representing plastid DNA were not present in the PacBio assembly. However, a BLAST search of a BGISeq-only assembly generated with SPAdes (v3.14.0) [120] against the *Nitzschia* Nitz4 plastid (GenBank accession: MG273660.1) [26] identified three contigs with a total size of 59 kb and a GC content of 25%, which most likely represent the plastid genome.

## Gene prediction and functional annotation

Genome annotation of protein-coding genes for the nuclear genome was performed using the MAKER (v3.01.03) [121] and BRAKER2 pipelines (v2.1.2) [122]. Predicted proteins from the diatoms *Fistulifera solaris* (GenBank accession: GCA_002217885.1) [123], *Pseudo-nitzschia multistriata* (GenBank accession: GCA_900660405.1) [124], *Phaeodactylum tricornutum* (GenBank accession: GCF_000150955.2) [38], *Fragilariopsis cylindrus* (GenBank accession: GCA_001750085.1) [39], *Chaetoceros tenuissimus* (GenBank accession: GCA_021927905.1) [125], *Fragilaria crotonensis* (GenBank accession: GCA_022925895.1) [126], and *Nitzschia inconspicua* (GenBank accession: GCA_019154785.2) [41] were used as inputs for both pipelines. For MAKER, the *N*. sing1 transcriptome assembled from two RNA-seq datasets with Trinity (v.2.4.0) [127] was also included.

Two independent annotations were generated using BRAKER2. The first set of annotations (BRAKER2 parameter: --etpmode) was generated based on the diatom protein datasets used as input and the BAM files of the two RNA-seq datasets mapped to the *N*. sing1 genome assembly with HISAT2 (v2.2.1) [128]. The second annotation set merges the results from two BRAKER2 runs (using TSEBRA v1.0.3) [129], which used either the protein or RNA-seq datasets as input. To derive at a final annotation set, all three gene model predictions from MAKER and BRAKER2 were combined using a custom Perl script. To assess the quality of the protein annotation, BUSCO (v5.5.0) analysis was performed using datasets from the eukaryote (eukaryote_odb10) and stramenopiles lineage (stramenopiles_odb10) (S1 Table). Putative mitochondria and plastid contigs were annotated using Prokka (v1.10) [130].

Functional annotation of *N*. sing1 protein-coding genes was performed by searching for conserved protein domains using hmmscan (Parameters: --domtblout mode, domain i-evalue $< 1.0E^{-03}$) from HMMER (v3.3.2) [131] against the Pfam protein database [132]. The signal peptide for each protein was predicted using a combination of Phobius [133,134] and SignalP6.0 [135]. Annotation of CAZymes was also performed with hmmscan (--domtblout mode, domain i-evalue $< 1.0E^{-10}$) using the dbCAN2 HMM database (HMMdb-V11) [136]. Proteins were assigned a CAZy family using the top EC hit with the lowest e-value. Heatmaps depicting the proportion of CAZymes in each genome were clustered using the Bray–Curtis dissimilarity measure. Scores for each EC/CBM class were calculated by normalizing the count against the total number of CAZyme within each species (Figs 1A and S2).

### *N*. sing1 alginate lyase nomenclature

ALY genes are numbered according to their position on contigs ordered from the largest to smallest. For proteins with single ALY and/or CBM32 domains, they are referred to as 'A' or 'C', respectively. For example, the A domain of ALY1 is referred to as ALY1-A. In cases where proteins encode more than one A- or C-domain, the domains are numbered by type according to their position from N- to C-terminus. For example, the third A domain of the $A^nC^n$ family protein $ALY48^{A5C5}$ is referred to as $ALY48^{A5C5}$-A3. TARP repeat sequences are numbered corresponding to the ALY gene from which they originate. For example, the TARP repeat from ALY1 is designated TARP1. In cases where TARP repeats are identical between ALY genes, they are identified by the first ALY gene in which they occur.

### Phylogenetic and sequence analysis

To identify the closest relatives to *N*. sing1 ALYs, BLASTP searches against the NCBI non-redundant database (nr) were performed with all 91 *N*. sing1 ALY sequences (BLASTP parameters: max target sequences 500, e-value threshold 0.05, word size 5, matrix: BLOSUM62, gap extension cost: 11, gap opening cost, 1, all other parameters default). BLASTP hits for all ALY query sequences were combined into a single dataset containing 27,230 unique sequences. Hits were then binned into 10 groups based on bit scores (interval of 50). Twenty-five sequences from each bin were then randomly sampled. For the phylogenetic analysis, sequences with ambiguous genera/species assignment or were not labeled as ALYs/polysaccharide lyases were removed. In addition, only one sequence was kept if multiple sequences from the same species (i.e., putative paralogs) were sampled. In this manner, we analyzed representative sequences from the full BLASTP output. Multiple sequence alignment was prepared with MAFFT (v7.490) [137] using the L-INS-I iterative refinement method and trimmed with trimAl (v1.4.rev15) (-gappyout option) [138]. Maximum likelihood phylogenetic trees were constructed using IQ-TREE2 (multicore v2.2.6) (Parameters: -b 1000 (bootstrap replicates)) [139,140] (Fig 1C).

To investigate the relationship between *N*. sing1 ALY genes, nucleotide sequences from all A-domains were compared and used to construct maximum likelihood phylogenetic trees (Figs 2A and S3). For proteins with multiple A-domains, each domain was included as separate sequences in the tree. Multiple sequence alignment and phylogenetic analysis were performed as described above. TARP repeat sequences (Figs 6 and S17 and S18) were defined in all CA family proteins using the C- and A-domains as boundaries. Multiple sequence alignment of TARP repeat sequences was performed with MAFFT (v7.490) using the E-INS-I iterative refinement method and trimmed with trimAl (v1.4.rev15) (-gappyout option). Maximum likelihood phylogenetic trees were constructed using IQ-TREE2 (multicore v2.2.6) (Parameters: -b 1000 (bootstrap replicates) (Figs 6A and S17B). For the alignment in S17B Fig, sequences were manually aligned to show TARP repeat expansions. Disorder in TARP repeat regions were predicted using IUPred3 [141] (S17A Fig). Visualization of sequence alignments at key A-domain catalytic regions was made using ESPript3.0 [142] (S10 Fig), while phylogenetic trees were visualized using iTOL (v6.9) [143].

### Detection of alginate metabolism orthologs

Candidate alginate metabolism genes in *N*. sing1 and select diatom species were detected using a curated set of alginate metabolism proteins (S5 Table) known in bacteria and some eukaryotes. These sequences were used to query against diatom proteins by BLASTP (BLASTP parameters: max target sequences 500, e-value threshold 0.05, word size 5, matrix: BLOSUM62, gap extension cost: 11, gap opening cost, 1, all other parameters default) and unique

hits above the e-value threshold ($1E^{-03}$) were counted for each species. To identify likely ortho-logs, MMSeqs2 (v15.6f452) [144] was used to perform a reciprocal best hit analysis (easy-rbh) between the diatom proteins and the curated protein set.

## Protein structural prediction

Structural predictions for A-domains were made using AlphaFold2 (ColabFold v1.5.5; Alpha-Fold2 using MMseqs2) with default parameters [145] (Figs 4C and S11). For ALY48$^{A5C5}$, the first A-domain (ALY48$^{A5C5}$-A1) was used for the prediction. The A-domain of AlyB from *V. splendidus* was obtained from PDB (PDB ID: 5ZU5) [55]. All visualizations were made and captured with PyMOL (v3.0.0).

## Analysis of ALY loci sequence homology

ALY loci were defined based on ALY gene clusters found on the *N.* sing1 genome assembly. Thirty ALY gene loci were manually assigned in total, comprising 91 ALY genes. To analyze these ALY loci, sequences were extracted with 2.5 kb extensions (flanks) on each end or until the ends of the contig for loci situated near contig terminals. Loci sequences were self-aligned using the Dotter software (v4.28) [146]. Homology between ALY genes and intergenic sequences within a locus was identified visually based on this output (S4–S8 Figs). Sequence homology for flank, genic and intergenic regions was measured through sequence percent similarity using the Needleman–Wunsch global alignment algorithm with modified param-eters (Figs 3C and S9). Specifically, pairwise comparisons between these regions were made for each locus using the Biostrings package (v2.74.0) in R (v4.2.2) [147]. Gap opening and extension penalties were increased (gapOpening = 25, gapExtension = 10) to obtain align-ments with contiguous regions of high sequence similarity. End gap penalties were omitted to account for differences in lengths (Parameter: type = "overlap"), and the EDNAFULL sub-stitution matrix was used. Percent similarity scores were calculated using the "PID3" formula (100 * (identical positions)/(length shorter sequence)) included in the Biostrings package.

## DNA amplification and gene cloning

Representative full-length ALY genes (from *N.* sing1 and *Vibrio* sp.) and TARP repeat regions (from *N.* sing1) were synthesized as *NdeI-BamHI* fragments in pET15b (GenScript). To express and purify individual A- and C-domains from *E. coli*, domain sequences were sub-cloned into a modified pET15b vector containing a C-terminal TEV-6xHIS sequence. To express and purify proteins as N-terminal mCherry or mGFP-fusions, a pET15b vector containing mCherry- or mGFP-TEV-6xHIS was used. To purify TARP-mCherry fusions, the mCherry gene was subcloned into the synthesized pET15b::6xHIS-*NdeI*-TARP-*BamHI* vector after the *BamHI* site. To assess the trafficking of signal peptides and TMDs, the predicted signal peptide from *N.* sing1 ALY49$^{A3}$ (Ref: Sing2_08162) (SP$^{49}$), *Vibrio casei* ALY (GenBank accession: WP_244913617.1) (SP$^{V.casei-ALY}$), and the human luminal protein BiP (SP$^{BiP}$) (Gen-bank accession: NP_005338.1) was appended to the N-terminal end of a sfGFP or mCherry insert by PCR. The C-terminus of these inserts was also appended with either the ER retention signal KDEL (for SP$^{BiP}$-sfGFP and SP$^{V.casei-ALY}$-sfGFP) or the predicted TMD from ALY49$^{A3}$ (for SP$^{49}$). For expression in HeLa cells, constructs were sub-cloned into the pcDNA3.1 + vector between the *NheI* and *XbaI* restriction sites. All constructs were produced through standard molecular biology techniques. PCR amplification was performed using the KAPA HiFi PCR Kit (Roche Diagnostics, 7958838001) and cloning was performed using the In-Fusion HD Cloning Kit (Takara Bio, 639650). All accessions, primers and vectors used in this study can be found in S6 Table.

## Protein expression and purification

All recombinant proteins were expressed in BL21 DE3 *E. coli* and purified as previously described [148] with some modifications. Briefly, overnight grown cultures were diluted 1:10 into fresh LB media (supplemented with 100 μg/mL ampicillin) at 37 °C and expression was induced with 1 mM isopropylthiogalactoside (IPTG) once the culture had reached an $OD_{600}$ of 0.6–0.8. Expression was carried out for 16–18 h at 16 °C (for ALY48$^{A5C5}$-A1) or 25 °C with shaking at 225 rpm. Cultures were then harvested, washed and cell pellets frozen at −80 °C. Cell pellets were resuspended in five volumes of lysis buffer and lysed by sonication on ice. Insoluble material was separated by centrifugation and the supernatant fraction was incubated with HisPur Ni-NTA resin (Thermo Scientific, 88222). Bound resin was washed three times before elution in lysis buffer containing 500 mM imidazole. Protein fractions were then pooled and dialyzed overnight in TBS (10 mM Tris, 150 mM NaCl) at 4 °C using 3,500 MWCO dialysis tubings (Spectrum Laboratories, 25219-041). Following dialysis, proteins were concentrated using 3,000 MWCO centrifugal filters (Merck, UFC500396) and protein concentrations were quantified by Bradford Assay (Bio-Rad, #5000006). For purification of CBM32-mCherry recombinant proteins, cell lysis and elution were performed in a denaturing buffer as previously described [148]. Protein re-folding was performed by dialysis in denaturing buffers containing decreasing levels of urea (4M, 2M, and 1M) for an hour each before exchanging into TBS overnight.

## Alginate liquefaction assays

Alginate liquefaction-vortex assays (Fig 4A) were performed using crude lysates of *E. coli* BL21 DE3 harboring ALY pET15b expression plasmids. Alginate gels were prepared by adding 20 mL 2% (w/v) medium-viscosity sodium alginate (Sigma-Aldrich, A2033) to 20 mL TBS-$Ca^{2+}$ buffer (10 mM Tris, 150 mM NaCl, 2 mM $CaCl_2$) with mixing. The resulting solution was briefly centrifuged (1,000 × *g*, 5 min) to create a uniform gel. To set up the assay, alginate gel was pipetted into a 15 mL polystyrene tube (Falcon, Corning) until the 2 mL mark and 80 μL cell lysate was added onto the gel surface. Reactions were incubated at 30 °C overnight. To quantify gel liquefaction, tubes were vortexed from the bottom for 5 s at 1.5 speed (on touch) using a vortex mixer (Vortex-Genie 2 SI-0236, Scientific Industries Inc) and the height of liquid displacement was marked and measured manually. All assays were performed in triplicate.

## Alginate lyase enzyme activity assays

ALY activity was determined spectrophotometrically at 235 nm by following the formation of 4,5-unsaturated bonds at the non-reducing end [149,150] (Fig 4B). Purified recombinant A-domains were added at a final concentration of 0.1 μM to an ALY enzyme activity buffer (0.2% (w/v) low-viscosity sodium alginate, 100 mM Tris, 2 mM $CaCl_2$) at a reaction volume of 200 μL. Purified mCherry-6xHIS and a commercial ALY (Sigma-Aldrich, A1603) serve as negative and positive controls, respectively. TBS was used as an additional negative control. Assays were prepared in triplicate in a 96-well UV-transparent microplate (Corning) and measurements were taken at 25 °C for 5 min (every 5 s interval) using a Tecan Spark Multimode Microplate Reader (Tecan Inc).

## Transient transfection of HeLa cells

To assess the trafficking of signal peptides, vectors encoding the signal peptide for *V. casei* or a positive control consisting of the signal peptide from the human luminal protein BiP were transfected into HeLa mammalian cells and imaged as previously described [151] (Fig 1D). HeLa cells were transfected with plasmids using the lipofectamine 3,000 transfection reagent

(Invitrogen, L3000001) and cultured for 48 h. Imaging was performed on the SP8 Inverted gSTED confocal microscope (Leica Microsystems) fitted with a 100× objective (NA: 1.4).

To monitor the trafficking signals derived from ALY49[A3], the SP[49]-mCherry-TMD[49] construct was co-transfected with the ER marker SP[BiP]-sfGFP-KDEL. Images were taken at 24 and 72 h post-transfection. For imaging, cells were cultured and transfected inside a 35 mm petri dish with a 20 mm glass bottom (VMR, 734-2906). Lysotracker-DND-26 (Invitrogen, L7526) was used to visualize lysosomes in transfected HeLa cells at a final concentration of 50 nM (Fig 5F). Lysotracker was added directly to the medium in the dish and incubated for 15 min at 37 °C. The probe-containing medium was then removed and replaced with PBS prior to imaging.

## Vacuole staining and fluorescence measurement

Fluorescent alginate moieties that appear upon growth of *N.* sing1 (Fig 5A and 5C) were visualized using epifluorescence and a CFP filter set (excitation 434 nm, emission 479 nm). Diatom vacuoles were stained using the CellTracker Blue CMAC dye (7-amino-4-chloromethylcoumarin) (Invitrogen, C2110) at a working concentration of 100 μM (Figs 5C and S16A). CMAC was added directly into the liquid culture medium and incubated at room temperature for 15 minutes. Cells were washed with an equal volume of SSW medium three times before imaging. To obtain and compare the emission spectra between vacuoles from alginate-grown diatoms (3 d) and the culture supernatant (4 d), samples were excited at 405 nm and scanned between 410–710 nm (step width: 20 nm). Intensities were normalized between 0 and 1. Data shown is derived from three independent scans (Fig 5E). Absorbance wavescans of diatom and *Vibrio* sp. culture media were performed using a Tecan Spark Multimode Microplate Reader (Tecan Inc) (Fig 5B).

## Alginate substrates

The alginates purchased from Sigma-Aldrich (A2033) and Pronova (42000101; 42000301) were analyzed using the standard protocol outlined by ASTM (American Society for Testing and Materials) method F2259-10 (2012) to determine their chemical composition, including sequential parameters, via NMR spectroscopy. Additionally, the molecular weight of the alginates was determined using SEC-MALS following the standard protocol provided by ASTM method F2605-08. For both methods, the analytical protocol for alginate [152] has been previously described. Poly-M was obtained from an epimerase-negative AlgG mutant of *Pseudomonas fluorescens* [152]. Poly-G and poly-MG were prepared as previously described [153]. Hydrolyzed seaweed alginate was prepared using an alginate sample extracted from *Laminaria hyperborea* leaf and subjected to stepwise acid hydrolysis to achieve an approximate degree of polymerization of 30. Initially, 1% (w/v) alginate was dissolved in ion-free water, with the pH adjusted to 5.6 using 0.1 M HCl, followed by hydrolysis at 95 °C for 1 h. Subsequently, the solution was cooled to room temperature in a water bath, and the pH was adjusted to 3.8. The hydrolysis was then continued at 95 °C for 50 min. After cooling again to room temperature in a water bath, the solution was neutralized to pH 6.8–7.5 using 0.1 M NaOH and then lyophilized.

## Alginate-binding assays

For fluorescence microscopy of alginate hydrogel binding (Figs 6B and S18B), the alginate hydrogel was prepared by mixing 5 mL 1% (w/v) medium-viscosity sodium alginate with 0.5 mL 1 M CaCl₂, adjusting the volume to 50 mL with sterile water and followed by centrifugation (3,000 × *g*, 4 °C, 15 min). The resulting pellet was resuspended in 25 mL TBS-Ca²⁺. Separately, TARP-mCh and ALY77-C-mGFP proteins were mixed with 2 mg/mL BSA in

TBS-Ca$^{2+}$ to a final concentration of 4 μM for each protein and incubated for 15 min at room temperature. The protein solution was then centrifuged (21,130 × $g$, 4 °C, 10 min) to remove any precipitates, mixed with alginate hydrogel in a 1:1 ratio and incubated at room temperature for 30 min. Following this, the hydrogel-protein solution was centrifuged (21,130 × $g$, 4 °C, 5 min) and the resulting pellet washed twice with TBS-Ca$^{2+}$. 5 μL of the solution was used for fluorescence microscopy (Olympus BX51, Olympus). mCherry-6xHIS was used as a negative control. For fluorescence microscopy of soluble alginate binding (Fig 6C), the recombinant protein was mixed with an alginate solution (0.3% medium-viscosity sodium alginate, 1 mg/mL BSA, TBS) to a final concentration of 4 μM, vortexed and diluted 64-fold in TBS before imaging (Olympus BX51, Olympus).

For alginate pelleting assays (S19A Fig), the recombinant protein was mixed with 1 mg/mL BSA to a final concentration of 4 μM and incubated for 15 min at room temperature. The solution was then centrifuged at full speed (21,130 × $g$, 4 °C, 10 min) and the supernatant mixed with alginate hydrogel (prepared as described above) in a 1:1 ratio (Total (T) fraction). Supernatant (S) and Pellet (P) fractions were obtained by centrifugation (21,130 × $g$, 4 °C, 10 min). Equal volumes of each fraction were then analyzed by SDS-PAGE (12% polyacrylamide gel). mCherry-6xHIS was used as a negative control.

FRAP (Fluorescence recovery after photobleaching) was used to compare the binding of C-domains from *N.* sing1 ALY77 (ALY77-C-mCh) and *Vibrio hyugaensis* (C$^{Vibrio}$-mCh) (S19B Fig). Alginate binding was performed as described in the pelleting assay, and samples were imaged using an Olympus FV3000 Inverted confocal system (Olympus) at 40× magnification (N.A.: 1.4). Four frames were taken before samples were bleached with 100% laser power at maximum speed for one frame (fifth frame) before recovery (frames taken every 2 s for up to 300 frames). FRAP analysis was performed using the cell-Sens software (Olympus). Normalization was carried out against frames 1–4. mCherry-6xHIS was used as a negative control.

Alginate binding was also assessed with gel filtration FPLC (ÄKTA Pure FPLC system, Cytiva) (Fig 6C). TARP-mCherry recombinant protein was mixed with 0.1% low-viscosity sodium alginate to a final concentration of 4 μM and incubated in TBS+ (10 mM Tris, 200 mM NaCl) for 15 min prior to sample injection. Mixtures were separated in TBS+ through a Superdex 200 Increase 10/300 GL (GE Healthcare, 28990944) column. Protein peaks were monitored at 260 and 587 nm.

### *Sargassum* staining

*Sargassum* material was collected at low tide from the intertidal zone on Sentosa island, Singapore (latitude 1.259895, longitude 103.810843; Singapore National Parks Board permit no. NP/RP20-016). *Sargassum* fronds were washed in SSW medium and stored in 70% ethanol to extract and remove the chlorophyll. Once coloration was sufficiently lost, 1 × 1 cm$^2$ were cut from the edge of the frond along the sagittal plane to produce thin strips of tissues. To minimize non-specific binding, strips were blocked with an SSW-based buffer containing 10 mg/mL BSA and 0.1% Tween 20 for 30 min. After blocking, tissues were washed with SSW and incubated with the recombinant proteins (4 μM) for 1 h. Following this, strips were washed in 200 μL SSW three times for 10 min each. The tissue was imaged using fluorescence microscopy (Olympus BX51, Olympus) at 100× magnification (S19C Fig).

### NMR analysis and time-resolved NMR

NMR spectra were acquired at 25 °C on a BRUKER AVIIIHD 800 MHz or Bruker AVIIIHD 600 MHz spectrometer (Bruker BioSpin AG, Fällanden, Switzerland) both equipped with a 5 mm z-gradient CP-TCI (H/C/N) cryogenic probe. The chemical shift was referenced to

residual water signal $^1$H: 4.75 ppm and the $^{13}$C chemical shift was referenced indirectly to water, based on the absolute frequency ratios [154]. Spectra were recorded, processed, and analyzed using TopSpin 3.5 or 4.1.4 (Bruker BioSpin). Reactions were run in 200 μL 20 mM HEPES pH 7.5, 25 mM NaCl, 2 mM $CaCl_2$ in $D_2O$ (D, 99.9%) in 3 mm LabScape Stream NMR tubes (Bruker LabScape).

For time-resolved experiments, 10 mg/mL seaweed alginate, poly-M, poly-MG or poly-G substrate (S7 Table) was dissolved in the HEPES buffer. 1D proton spectrum with water suppression (noesygppr1d) was recorded at 25 °C to verify the sample's integrity before the time-resolved NMR experiment. The reaction was started by adding 1–2 μL of ALY7-A (50 μM), ALY48$^{A5C5}$-A1 (60 μM), ALY58-A (50 μM) or ALY77-A (50 μM) to the preheated substrate and mixed by inverting the sample a few times. The sample was then immediately re-inserted into the NMR instrument and the experiment started. The recorded spectrum is a pseudo-2D type experiment recording a 1D proton NMR spectrum (based on noesygppr1d) every 5 min with a total of 64 time points (total experiment time 5 h 20 min). After each time-resolved experiment, a $^1$H-$^{13}$C HSQC (heteronuclear single quantum coherence; hsqcetgpsisp2.2) spectrum with multiplicity editing was recorded. Signals were assigned based on previously published assignments [155–158] (Figs 4D and 4E and S12–S15). The preferred mode of action for the lyases (Fig 4F) was evaluated by integrating the peaks in the $^1$H-$^{13}$C HSQC after the reaction was completed. ΔGβ, ΔGα/poly, and ΔMβ/poly serve as markers of endolytic activity, while DEH and DHF serve as exolytic markers. ΔGα dimer was calculated using the $\alpha/\beta$ ratio (ΔGα/Gβ = 0.27) [153] due to overlapping signals.

## Supporting information

**S1 Fig. GenomeScope profile shows that *N*. sing1 is likely to be diploid.** 21-mers (*k*-mers) were counted using the tool Jellyfish and analyzed with GenomeScope. The presence of two major peaks observed in the GenomeScope profile suggests that *N*. sing1 is diploid. The data underlying this figure can be found in S1 Data.
(TIF)

**S2 Fig. Complete carbohydrate-active enzyme (CAZyme) analysis.** The heatmap displays the proportion of genes annotated with CAZyme family domains (CAZyme family genes, magenta scale) or carbohydrate-binding domains (CBM-containing genes, green scale) as a percentage of the total CAZyme genes annotated in each species. Red arrows denote families that may have undergone expansion in the apochlorotic or *N*. sing1 lineage. The data underlying this figure can be found in S2 Data.
(TIF)

**S3 Fig. Maximum likelihood phylogenetic tree of *N*. sing1 ALY alginate lyase domains.** The phylogenetic tree (bootstrap replicates = 1000) was constructed using the nucleotide sequences of all *N*. sing1 ALY A-domains. Bootstrap support for nodes is summarized in the legend. Tree scale = 0.1. Domain organization of the ALY gene is illustrated next to each leaf. The domain corresponding to the leaf is identified by a black border. Domains belonging to the CCA or A subfamily ALYs are marked with a red asterisk. The presence/absence of signal peptide and transmembrane domain (TMD) in the ALY gene each A-domain belongs to, along with whether the domain contains mutations at key catalytic sites are summarized on the right. Clades highlighted in yellow, blue and red fall under the CA, A$^{n-TMD}$ and A$^n$C$^n$ families, respectively. The node giving rise to the derived ALY families (A$^{n-TMD}$ and A$^n$C$^n$) is marked with a black arrow. The data underlying this figure can be found in S5 Data.
(TIF)

**S4 Fig. Sequence dot plots of ALY loci in the *N*. sing1 genome (Loci 1–6).** Self-alignment of ALY loci 1–6 nucleotide sequence (± 2.5 kb) reveals patterns of DNA homology within each locus. The gene structure of the ALY locus is shown along both axes. Protein domains are labeled according to the legend. Dashed dark blue and red boxes identify homology between genic and intergenic regions of the locus, respectively.
(TIF)

**S5 Fig. Sequence dot plots of ALY loci in the *N*. sing1 genome (Loci 7–12).** Self-alignment of ALY loci 7–12 nucleotide sequence (± 2.5 kb) reveals patterns of DNA homology within each locus. The gene structure of the ALY locus is shown along both axes. Protein domains are labeled according to the legend. Dashed dark blue and red boxes identify homology between genic and intergenic regions of the locus, respectively.
(TIF)

**S6 Fig. Sequence dot plots of ALY loci in the *N*. sing1 genome (Loci 13–18).** Self-alignment of ALY loci 13–18 nucleotide sequence (± 2.5 kb) reveals patterns of DNA homology within each locus. The gene structure of the ALY locus is shown along both axes. Protein domains are labeled according to the legend. Dashed dark blue and red boxes identify homology between genic and intergenic regions of the locus, respectively.
(TIF)

**S7 Fig. Sequence dot plots of ALY loci in the *N*. sing1 genome (Loci 19–24).** Self-alignment of ALY loci 19–24 nucleotide sequence (± 2.5 kb) reveals patterns of DNA homology within each locus. The gene structure of the ALY locus is shown along both axes. Protein domains are labeled according to the legend. Dashed dark blue and red boxes identify homology between genic and intergenic regions of the locus, respectively.
(TIF)

**S8 Fig. Sequence dot plots of ALY loci in the *N*. sing1 genome (Loci 25–30).** Self-alignment of ALY loci 25–30 nucleotide sequence (± 2.5 kb) reveals patterns of DNA homology within each locus. The gene structure of the ALY locus is shown along both axes. Protein domains are labeled according to the legend. Dashed dark blue and red boxes identify homology between genic and intergenic regions of the locus, respectively.
(TIF)

**S9 Fig. Sequence homology between intergenic regions in *N*. sing1 ALY loci.** (Top) The cartoon illustrates how unequal crossing-over at a locus with two tandem genes can lead to both duplication of the gene and intergenic region. (Bottom) Self-comparisons of intergenic sequences within all *N*. sing1 ALY loci (loci 1–30). The percent sequence identity for each pairwise comparison is shown as a matrix plot. IGR, intergenic region; LF, left flank; RF, right flank. The data underlying this figure can be found in S6 Data.
(TIF)

**S10 Fig. Protein sequence alignment of PL7 catalytic regions in $A^{n-TMD}$ and $A^nC^n$ family ALY proteins (A-domain).** Protein sequence alignment of $A^{n-TMD}$ and $A^nC^n$ family ALYs (A-domain catalytic regions; A–C) in *N*. sing1. Conserved catalytic residues are highlighted in yellow (A: RXE(L/V)R; B: Q(I/V)H; and C: YFKXGXYXQ). The red arrowheads denote A-domains with substitutions at key catalytic residues in at least one of these three catalytic regions.
(TIF)

**S11 Fig. Structural predictions of *N*. sing1 ALY proteins.** (A) (Left to right) Cartoon representations of the protein crystal structure of AlyB alginate lyase domain (*V. splendidus*; PDB

5ZU5) and AlphaFold2 predictions of ALY77 and ALY48$^{A5C5}$ A-domains (*N. sing1*; ALY77-A and ALY48$^{A5C5}$-A1). Top and bottom rows show two different views along the catalytic cleft. Putative catalytic residues (R/Q/H/Y) are colored in magenta, while the 15-bp insertion predicted to contribute to exolytic activity is colored in green. Magenta arrows and circles indicate the likely binding site of alginate molecules along the catalytic cleft. The data underlying this figure can be found at https://zenodo.org/records/14793551. (**B**) Predicted aligned error (PAE) plots of the two AlphaFold2-predicted structures in (A) show the confidence in the relative position of two residues within the predicted structure. The data underlying this figure can be found in S8 Data. (**C**) Predicted local distance difference test (pLDDT) scores for each residue of the two AlphaFold2-predicted structures in (A), which signifies the confidence scores of the prediction. The data underlying this figure can be found in S8 Data. (**D**) Magnified view of the alginate lyase catalytic groove (displayed as surface representations). Note that the loop insertion in ALY48$^{A5C5}$-A1 appears to block the groove.
(TIF)

**S12 Fig. Activities of recombinant *N. sing1* ALY48$^{A5C5}$-A1 on different alginate substrates.** Anomeric and Δ H/C-4 regions of $^1$H-$^{13}$C HSQC of (1) Seaweed alginate (F$_G$ 0.46; DP ~30) (top left panel), (2) Poly-M (DP ~30) (top right panel), (3) Poly-MG (DP ~26) (bottom left panel), and (4) Poly-G (DP ~26) (bottom right panel) treated with ALY48$^{A5C5}$-A1 after 5 h 20 min (in 20 mM HEPES (pH 7.5), 25 mM NaCl, 2 mM CaCl$_2$ in D$_2$O (D, 99.9%) recorded on a 600 or 800 MHz instrument at 25 °C. The inlay panels show $^1$H time-resolved spectrum of the ΔH/C-4 region over the reaction period. M: Mannuronate; G: Guluronate; Δ: ΔH/C-4 of 4,5-unsaturated 4-deoxy-L-erythro-hex-4-enepyranosyluronate; Δ: ΔH/C-1 (anomeric signal) of 4,5-unsaturated 4-deoxy-L-erythro-hex-4-enepyranosyluronate; DEH: 4-deoxy-L-erythro-5-hexulosuronate hydrate; DHF: two epimers 4-deoxy-D-manno-(5*S*)-hexulofuranosidonate hydrate and 4-deoxy-D-manno-(5*R*)-hexulofuranosidonate hydrate; α/β: M/G at reducing ends of alginate residue; poly: M/G in a polymer; GM-/MG-: Alternating GM/MG polymer. Underlined labels indicate the residue with the anomeric H/C-1 giving rise to the signal, and -xx- indicates signals within a polysaccharide chain. The data underlying this figure can be found at https://zenodo.org/records/14410447.
(TIF)

**S13 Fig. Activities of recombinant *N. sing1* ALY7-A on different alginate substrates.** Anomeric and ΔH/C-4 regions of $^1$H-$^{13}$C HSQC of (1) Seaweed alginate (F$_G$ 0.46; DP ~30) (top left panel), (2) Poly-M (DP ~30) (top right panel), (3) Poly-MG (DP ~26) (bottom left panel), and (4) Poly-G (DP ~26) (bottom right panel) treated with ALY7-A after 5 h 20 min (in 20 mM HEPES (pH 7.5), 25 mM NaCl, 2 mM CaCl$_2$ in D$_2$O (D, 99.9%) recorded on a 600 or 800 MHz instrument at 25 °C. The inlay panels show $^1$H time-resolved spectrum of the ΔH/C-4 region over the reaction period. The inlay panel for (2) Poly-M is omitted due to the absence of activity. M: Mannuronate; G: Guluronate; Δ: ΔH/C-4 of 4,5-unsaturated 4-deoxy-L-erythro-hex-4-enepyranosyluronate; Δ: ΔH/C-1 (anomeric signal) of 4,5-unsaturated 4-deoxy-L-erythro-hex-4-enepyranosyluronate; DEH: 4-deoxy-L-erythro-5-hexulosuronate hydrate; DHF: two epimers 4-deoxy-D-manno-(5*S*)-hexulofuranosidonate hydrate and 4-deoxy-D-manno-(5*R*)-hexulofuranosidonate hydrate; α/β: M/G at reducing ends of alginate residue; poly: M/G in a polymer; GM-/MG-: Alternating GM/MG polymer. Underlined labels indicate the residue with the anomeric H/C-1 giving rise to the signal, and -xx- indicates signals within a polysaccharide chain. The data underlying this figure can be found at https://zenodo.org/records/14410447.
(TIF)

**S14 Fig. Activities of recombinant *N.* sing1 ALY58-A on different alginate substrates.** Anomeric and ΔH/C-4 regions of $^1$H-$^{13}$C HSQC of (1) Seaweed alginate ($F_G$ 0.46; DP ~30) (top left panel), (2) Poly-M (DP ~30) (top right panel), (3) Poly-MG (DP ~26) (bottom left panel), and (4) Poly-G (DP ~26) (bottom right panel) treated with ALY58-A after 5 h 20 min (in 20 mM HEPES (pH 7.5), 25 mM NaCl, 2 mM CaCl$_2$ in D$_2$O (D, 99.9%) recorded on a 600 or 800 MHz instrument at 25°C. The inlay panels show $^1$H time-resolved spectrum of the ΔH/C-4 region over the reaction period. The inlay panel for (2) Poly-M is omitted due to the absence of activity. M: Mannuronate; G: Guluronate; Δ: ΔH/C-4 of 4,5-unsaturated 4-deoxy-ʟ-erythro-hex-4-enepyranosyluronate; Δ: ΔH/C-1 (anomeric signal) of 4,5-unsaturated 4-deoxy-ʟ-erythro-hex-4-enepyranosyluronate; DEH: 4-deoxy-ʟ-erythro-5-hexulosuronate hydrate; DHF: two epimers 4-deoxy-ᴅ-manno-(5*S*)-hexulofuranosidonate hydrate and 4-deoxy-ᴅ-manno-(5*R*)-hexulofuranosidonate hydrate; α/β: M/G at reducing ends of alginate residue; poly: M/G in a polymer; GM-/MG-: Alternating GM/MG polymer. Underlined labels indicate the residue with the anomeric H/C-1 giving rise to the signal, and -xx- indicates signals within a polysaccharide chain. The data underlying this figure can be found at https://zenodo.org/records/14410447. (TIF)

**S15 Fig. Activities of recombinant *N.* sing1 ALY77-A on different alginate substrates.** Anomeric and ΔH/C-4 regions of $^1$H-$^{13}$C HSQC of (1) Seaweed alginate ($F_G$ 0.46; DP ~30) (top left panel), (2) Poly-M (DP ~30) (top right panel), (3) Poly-MG (DP ~26) (bottom left panel), and (4) Poly-G (DP ~26) (bottom right panel) treated with ALY77-A after 5 h 20 min (in 20 mM HEPES (pH 7.5), 25 mM NaCl, 2 mM CaCl$_2$ in D$_2$O (D, 99.9%) recorded on a 600 or 800 MHz instrument at 25 °C. The inlay panels show $^1$H time-resolved spectrum of the ΔH/C-4 region over the reaction period. The inlay panel for (2) Poly-M and (4) Poly-G is omitted due to the absence of activity and lock error during recording, respectively. M: Mannuronate; G: Guluronate; Δ: ΔH/C-4 of 4,5-unsaturated 4-deoxy-ʟ-erythro-hex-4-enepyranosyluronate; Δ: ΔH/C-1 (anomeric signal) of 4,5-unsaturated 4-deoxy-ʟ-erythro-hex-4-enepyranosyluronate; DEH: 4-deoxy-ʟ-erythro-5-hexulosuronate hydrate; DHF: two epimers 4-deoxy-ᴅ-manno-(5*S*)-hexulofuranosidonate hydrate and 4-deoxy-ᴅ-manno-(5*R*)-hexulofuranosidonate hydrate; α/β: M/G at reducing ends of alginate residue; poly: M/G in a polymer; GM-/MG-: Alternating GM/MG polymer. Underlined labels indicate the residue with the anomeric H/C-1 giving rise to the signal, and -xx- indicates signals within a polysaccharide chain. The data underlying this figure can be found at https://zenodo.org/records/14410447. (TIF)

**S16 Fig. Cell biology of diatom alginate metabolism.** (**A**) Vacuole staining in diatoms grown on dextrose and alginate. CMAC dye shows an accumulation of vacuoles in diatoms grown on alginate, but not on dextrose (*n* = 10). (**B**) Culture medium undergoes yellowing over time in *N.* sing1 cultures (middle, 72 h), but not in *Vibrio* sp. culture (bottom, 72 h). (TIF)

**S17 Fig. Sequence analysis of TARP repeat sequences.** (**A**) The heatmap shows the predicted disorder scores (IUPred3) for *N.* sing1 CA family ALY proteins (green/blue scale). Note that the disordered regions are variable in length and occur between the C- and A-domains. The black lines denote the boundary formed by the annotated C- and A-domains. The data underlying this figure can be found in S10 Data. (**B**) Maximum likelihood phylogenetic tree (bootstrap replicates = 1000) and protein sequence alignment of *N.* sing1 CA family ALY TARP repeat region. The alignment was manually generated to highlight how TARP repeats vary in tetrapeptide unit lengths between closely related ALYs.

TARP repeats are in bold and arginine residues are highlighted in red. The data underlying this figure can be found in S10 Data.
(TIF)

**S18 Fig. TARP repeat sequences bind to alginate hydrogels.** (**A**) The cartoon illustrates the set of TARP repeat regions synthesized as gene constructs. These constructs are named after the first ALY gene they are found in (e.g. TARP1 is found in ALY1). ALY genes with the TARP repeat regions are listed on the right. TARP repeat units are shaded in gray while the red line represents the arginine residue. (**B**) Double staining of alginate hydrogels with recombinant TARP-mCherry and ALY77-C-mGFP fusion proteins. $N_{TARP}$ denotes the number of TARP repeats present in the recombinant protein. Brighter signals are observed in recombinant proteins that contain more TARP repeats. Samples were imaged at 100× magnification (600 ms exposure) using the TRIT-C (mCherry; top row) and GFP (mGFP; bottom row) channels (0 low–800 high). For TARP32-mCh, imaging was performed using a shorter exposure (200 ms) due to signal saturation. Scale = 10 μm.
(TIF)

**S19 Fig. Alginate-binding assays for CBM32 and TARP repeat regions.** (**A**) SDS-PAGE of total (T), supernatant (soluble) (S) and pellet (insoluble) (P) fractions from an alginate pelleting assay using ALY77-C-mCh and $C^{Vibrio}$-mCh recombinant proteins. A protein band is observed in the pellet fraction for both C-domain fusion proteins but not mCherry, indicating that both C-domains bind to alginate. (**B**) Fluorescence Recovery After Photobleaching (FRAP). The recovery time for both C-domain fusion proteins following bleaching is slower than mCherry, indicating that both C-domains associate with the alginate gel. A stronger association for ALY77-C-mCh compared to $C^{Vibrio}$-mCh is observed. The data underlying this figure can be found in S12 Data. (**C**) *Sargassum* tissue binding assay using C-domain fusion proteins and TARP 9X-mCh recombinant protein. All three fusion proteins bound to the cell wall of *Sargassum* tissues, as seen from the signal when tissues were visualized under epifluorescence. No signal was observed in the mCherry control. Scale = 100 μm.
(TIF)

**S1 Table. Diatom genome assembly statistics.**
(XLSX)

**S2 Table. Putative telomeric repeats in the *N. sing1* assembly.**
(XLSX)

**S3 Table. CAZyme gene counts.**
(XLSX)

**S4 Table. PacBio long reads mapping to tandem ALY loci.**
(XLSX)

**S5 Table. Alginate metabolism orthologs in diatom species.**
(XLSX)

**S6 Table. Primers and vectors used in this study.**
(XLSX)

**S7 Table. Characteristics of alginates used in this study.**
(XLSX)

**S1 Data. Raw data for *N. sing1* GenomeScope profile.**
(XLSX)

**S2 Data. Raw data for CAZy analysis heatmaps.**
(XLSM)

**S3 Data. *N*. sing1 Alginate Lyase (ALY) gene and domain annotation.**
(XLSM)

**S4 Data. Maximum likelihood phylogenetic tree of *N*. sing1 CA family ALYs and closely related bacteria alginate lyases.**
(ZIP)

**S5 Data. Maximum likelihood phylogenetic tree of *N*. sing1 ALY genes (A-domains).**
(ZIP)

**S6 Data. Raw data for intergenic sequence homology matrix plots.**
(XLSX)

**S7 Data. Raw data for alginate gel liquefaction and alginate lyase enzyme activity assays.**
(XLSX)

**S8 Data. Raw data for AlphaFold2 PAE and pLDDT plots.**
(XLSX)

**S9 Data. Raw data for diatom cell biology experiments (Fig 5).**
(XLSX)

**S10 Data. Raw data for TARP region sequence analysis.**
(XLSX)

**S11 Data. Raw data for gel filtration chromatograms.**
(XLSX)

**S12 Data. Raw data for FRAP (C-domains).**
(XLSM)

## Acknowledgments

FLA is grateful for experimental assistance from Synnøve Strand Jacobsen and Alexander Mika Hannasvik. We thank Mirjam Czjzek and Antonia Monteiro for helpful discussions.

## Author contributions

**Conceptualization:** Gregory Jedd.

**Data curation:** Zeng Hao Lim, Minou Nowrousian.

**Formal analysis:** Zeng Hao Lim, Peng Zheng, Christopher Quek, Minou Nowrousian, Finn L. Aachmann.

**Funding acquisition:** Gregory Jedd.

**Investigation:** Zeng Hao Lim, Peng Zheng, Christopher Quek, Minou Nowrousian, Finn L. Aachmann, Gregory Jedd.

**Methodology:** Zeng Hao Lim, Peng Zheng, Christopher Quek, Minou Nowrousian, Finn L. Aachmann, Gregory Jedd.

**Supervision:** Gregory Jedd.

**Visualization:** Zeng Hao Lim, Peng Zheng, Christopher Quek, Finn L. Aachmann, Gregory Jedd.

**Writing – original draft:** Zeng Hao Lim, Gregory Jedd.

**Writing – review & editing:** Zeng Hao Lim, Peng Zheng, Christopher Quek, Minou Nowrousian, Finn L. Aachmann, Gregory Jedd.

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
