## [Editor Report · Decision Letter 0]

10 Sep 2024

Dear Dr Jedd, 

Thank you for submitting your manuscript entitled "Innovations in Alginate Catabolism Leading to Heterotrophy and Adaptive Radiation of Diatoms" for consideration as a Research Article by PLOS Biology.

Your manuscript has now been evaluated by the PLOS Biology editorial staff, as well as by an academic editor with relevant expertise, and I am writing to let you know that we would like to send your submission out for external peer review.

Once your full submission is complete, your paper will undergo a series of checks in preparation for peer review. After your manuscript has passed the checks it will be sent out for review. To provide the metadata for your submission, please Login to Editorial Manager (https://www.editorialmanager.com/pbiology) within two working days, i.e. by Sep 12 2024 11:59PM.

Kind regards,

Melissa

Melissa Vázquez Hernández, PhD

Associate Editor, PLOS Biology

on behalf of

Roland

Roland Roberts, PhD

Senior Editor

PLOS Biology

rroberts@plos.org

---

## [Decision Letter · Decision Letter 1]

8 Nov 2024

Dear Greg,

Thank you for your patience while your manuscript "Innovations in Alginate Catabolism Leading to Heterotrophy and Adaptive Evolution of Diatoms" was peer-reviewed at PLOS Biology. It has now been evaluated by the PLOS Biology editors, an Academic Editor with relevant expertise, and three independent reviewers.

Based on the reviews and our Academic Editor's assessment of your paper, we are likely to accept this manuscript for publication, provided you satisfactorily address the remaining points raised by the reviewers. Please also make sure to address the following data and other policy-related requests.

IMPORTANT - please attend to the following:

a) Please make your Title more explicit and appealing to a broader readership. We suggest "Gene transfer from bacteria enabled Nitzchia diatoms to switch from photosynthesis to heterotrophic consumption of brown algal polysaccharides"

b) Please address the reviewers' concerns.

c) Please address my Data Policy requests below; specifically, we need you to supply the numerical values underlying Figs 1A, 2A (treefile), 3B, 4AB, 5BDE, 6C, S1, S2, S3 (treefile), S4 (treefiles), S6, S7BC, S10, S12A, S14B, either as a supplementary data file or as a permanent DOI’d deposition.

d) Please cite the location of the data clearly in all relevant main and supplementary Figure legends, e.g. “The data underlying this Figure can be found in S1 Data” or “The data underlying this Figure can be found in https://zenodo.org/records/XXXXXXXX

e) Please make any custom code available, either as a supplementary file or as part of your data deposition.

We expect to receive your revised manuscript within two weeks. 

*Published Peer Review History*

*Press*

Sincerely,

Roli

Roland Roberts, PhD

Senior Editor

rroberts@plos.org

PLOS Biology

DATA POLICY:

Regardless of the method selected, please ensure that you provide the individual numerical values that underlie the summary data displayed in the following figure panels as they are essential for readers to assess your analysis and to reproduce it: Figs 1A, 2A (treefile), 3B, 4AB, 5BDE, 6C, S1, S2, S3 (treefile), S4 (treefiles), S6, S7BC, S10, S12A, S14B. NOTE: the numerical data provided should include all replicates AND the way in which the plotted mean and errors were derived (it should not present only the mean/average values).

CODE POLICY

We require the original, uncropped and minimally adjusted images supporting all blot and gel results reported in an article's figures or Supporting Information files. We will require these files before a manuscript can be accepted so please prepare and upload them now. Please carefully read our guidelines for how to prepare and upload this data: https://journals.plos.org/plosbiology/s/figures#loc-blot-and-gel-reporting-requirements

DATA NOT SHOWN?

REVIEWERS' COMMENTS:

Reviewer #1:

Review of 'Innovations in Alginate Catabolism Leading to Heterotrophy and Adaptive Evolution of Diatoms'

This excellent paper sequences the genome of a heterotrophic diatom that has recently lost photosynthetic function to become a heterotroph feeding off alginate provided by seaweeds. The aim of the work is to understand how this change in lifestyle was underpinned by genomic and functional adaptations. 

The authors produce a high-quality genome assembly of the diatom. Analysis of the genome identifies a highly duplicated gene family predicted to function in the breakdown of alginate. They then show this gene family was gained by HGT from marine bacteria and then subject to numerous rounds of gene duplication. They nicely demonstrate that this pattern of gene duplication is the product of gene transposition. 

They then depart into a series of experiments to investigate the function of these paralogue families. The authors confirm the function of the gene (alginate breakdown) but also demonstrate that multiple paralogue families have diversified into differing functions including binding and transportation of alginate. 

The quality in this paper is very high. They conduct a series of experiments to investigate a number of related questions that often arise from analysis of HGT events, for example they test the possibility that the [approximate] bacterial donor gene secretion signal peptide would function in a eukaryotic system. It does, and so they add considerable weight to the idea that the HGT they have observed would lead to a direct gain of function. 

I think this is a really good piece of work, few explorations of HGT include this attention to functional biology of the HGT family, I only have minor suggestions. 

1) I think the phylogenies showing the HGT from bacteria should be a primary figure not a supplementary figure.

2) Paragraph 298-315. When discussing co-localisation with lysosome. I don't doubt this data, but firstly here I think the authors should state in this section that this is assayed by co-localisation with Lyostracker (unless I missed something). This information is in the methods but should be stated here as well. I think the authors should also add a caveat that Lysotracker in diatoms can also stain a range of acidic vesicles, some associated with exoskeleton formation, so it is not 100% reliable for pure lyososomes organelles as understood for animal cella. Only a brief caveat is needed. 

3) Line 315 'which is consistent with the behaviour of a eukaryotic import receptor'. Please add a reference.

4) Very minor point - line 582-586. I had to reread this section several times to get a sense of how they processed the alignment sequence gather for their phylogenetic tree analysis. My suggestion is that the authors just reread and may be edit this section for clarity. 

Reviewer #2:

The manuscript by Lim et al. describes the discovery and analysis of alginate catabolism in a diatom (Nitzschia sing1) that has lost the ability to carry out photosynthesis. The paper is well-written, combines multiple innovative techniques and analyses to document an initial horizontal gene transfer that was followed by multiple gene duplication events and enzyme neofunctionalization. I congratulate the authors for their extensive and well-described study. 

I have only minor suggestions.

Title - as written, the title implies that alginate catabolism is what led to the loss of photosynthesis by these diatoms. This possibility was not really the topic of the manuscript. I suggest that the authors modify the title to more accurately describe the findings of their paper.

Line 465. The authors state "Diatoms are likely to possess features that predispose them to a successful transition to obligate heterotrophy." If this was the case, then more diatom lineages would be expected to be obligate heterotrophs, which is not the case. I suggest the authors tone down this statement. 

Fig. 2A legend. I could not find the "shaded in gray" in figure that the authors indicate in the legend. 

Reviewer #3:

The study "Innovations in Alginate Catabolism Leading to Heterotrophy and Adaptive Evolution of Diatoms" by Lim et al. is a beautiful piece of work that reveals a fascinating biological story about how evolution is connected to the ecology and biochemistry of diatoms. It was a pleasure to read and will be relevant and exciting for others, too. Through genome sequencing and phylogenetic analyses, the authors show that the diatom Nitzschia sing1 acquired alginate metabolism through an ancient horizontal gene transfer (HGT) event from bacteria. This sequence then diversified within the diatom through gene duplication, resulting in 91 different homologs. This phenomenon is frequently observed in bacterial populations degrading chitin (Datta, 2016) or alginate (Hehemann, 2016), but the mechanism of evolution is HGT in bacteria, and the exact role of these different enzyme homologs remains largely unclear. By providing various types of enzyme activity assays (NMR, binding, imaging), the authors demonstrate functional differences between the diversified sequences and present compelling evidence that these sequences diversified to adapt to novel functions, such as exo-active mode and binding. For example, they identify a 15 bp insertion in some sequences that could convert endo- to exo-enzymes or, along with other mutations, enhance alginate binding. Overall, the study addresses a very important question about the evolution of carbohydrate metabolism, and all major conclusions by the authors are well-supported by high-quality data and experiments. I recommend publication with only one major edit concerning writing and data representation, which I am confident the authors can address.

My main point is that I am essentially lost between Figures 2 and 3 (lines 182-217) due to the overall data visualization, narrative in the text, and unclear scientific logic. I believe this section needs reworking in both the text and figures. Here are some specific points:

1. In Figures 1 and 2, the trees lack branch lengths, and Figure 2 does not have leaf labels, which makes it difficult to follow the authors' reasoning without referring to the tree in Subfigure 4. It would be beneficial to make the trees more reader-friendly by adding metadata, such as domain organization or activity data, directly onto the tree as a heatmap or similar.

2. Figure 3 is particularly hard to understand. I don't see the need for Panel A, which merely shows gene orientation; there are few inversions, so what am I supposed to take away from this? Panel B is intended to explain the logic of unequal crossing over, but I needed to read up on this to comprehend the illustration, as it is not explained in the main text.

3. The text section lacks a clear statement that these sequences evolve through duplication. I expect higher intra-sequence homology of A domains in AnCn sequences than between different sequences. While this is visible in Supplementary Figure 4, it is not clearly established in the main text or figures. The authors should clarify this point. There are other systems where additional protein domains evolve through recombination or HGT with other regions.

4. From Supplementary Figure 4, it appears that the endo-lyases are less diverse, while there is more rapid evolution/diversification in Atmd and AnCn proteins, which aligns with the overall hypothesis of adaptive radiation in these sequences. Why is this not mentioned in the main text?

5. In line 186, they state "orientation that conforms to duplication through unequal crossing over," but in line 201, they discuss transposition. These are different mechanisms of gene duplication, and it is unclear how the authors distinguish between the two scenarios given the data they have.

6. It is difficult to see in Supplementary Figure 4, but it seems that there is also a small amount of sequence divergence in different A domains within one AnCn protein. Why is this not discussed in more detail? Would it not make sense to analyze the rate of evolution within the A domains before examining the inter-gene regions? Additionally, a clear comparison of the evolution of the A domain versus the IG domain is lacking. What type of information can be learned from the similarity of the A domain, and why is it necessary to look at the IG domain? As I am not a geneticist, more explanations would help me follow the argument.

Minor points:

* Line 21: The phrase "manner of energy acquisition is unclear" is confusing; it states in the same sentence that it is a heterotroph. Please specify what is unclear.

* Line 111: Where was growth on seaweed polysaccharides shown?

* Lines 431-433: Are the monosaccharide catabolic genes known to metabolize acidic sugars in bacteria absent in this diatom?

* Line 456: Why was Sargassum chosen? In line 128, you mention it as an epiphyte of brown algae; which one? I had to refer back to reference 32 for clarification, but it would be better to include these details in the main text.

* Figure 2: I don't see the gray shaded area intended to highlight Antmd or AnCn domains.

---

## [Editor Report · Decision Letter 2]

27 Jan 2025

Dear Greg,

Thank you for the submission of your revised Research Article "Diatom heterotrophy on brown algal polysaccharides emerged through horizontal gene transfer, gene duplication and neofunctionalization" for publication in PLOS Biology. On behalf of my colleagues and the Academic Editor, Aoife McLysaght, I'm pleased to say that we can in principle accept your manuscript for publication, provided you address any remaining formatting and reporting issues. These will be detailed in an email you should receive within 2-3 business days from our colleagues in the journal operations team; no action is required from you until then. Please note that we will not be able to formally accept your manuscript and schedule it for publication until you have completed any requested changes.

N.B. As noted in a separate email, I've taken the liberty of including an active verb in your Title, as per our style guide. Als I should let you know that we don't use graphical abstracts (I saw that you included one) - feel free to include it as a "striking image" for our social media etc.

Sincerely,

Roli

Senior Editor

PLOS Biology

rroberts@plos.org